# Hadley cell expansion in CMIP6 models

Kevin M. Grise[1], Sean M. Davis[2]

[1]Department of Environmental Sciences, University of Virginia, Charlottesville, VA 22904, USA
[2]NOAA Earth System Research Laboratory Chemical Sciences Division, Boulder, CO 80305, USA

*Correspondence to*: Kevin M. Grise (kmg3r@virginia.edu)

**Abstract.** In response to increasing greenhouse gases, the subtropical edges of Earth's Hadley circulation shift poleward in global climate models. Recent studies have found that reanalysis trends in the Hadley cell edge over the past 30–40 years are within the range of trends simulated by Coupled Model Intercomparison Project Phase 5 (CMIP5) models, and have documented seasonal and hemispheric asymmetries in these trends. In this study, we evaluate whether these conclusions

hold for the newest generation of models (CMIP6). Overall, we find similar characteristics of Hadley cell expansion in CMIP5 and CMIP6 models. In both CMIP5 and CMIP6 models, the poleward shift of the Hadley cell edge in response to increasing greenhouse gases is 2–3 times larger in the Southern Hemisphere (SH), except during September–November. The trends from CMIP5 and CMIP6 models agree well with reanalyses, although prescribing observed coupled atmosphere-ocean variability allows the models to better capture reanalysis trends in the Northern Hemisphere (NH). We find two

notable differences between CMIP5 and CMIP6 models. First, while both CMIP5 and CMIP6 models contract the NH summertime Hadley circulation equatorward (particularly over the Pacific sector), this contraction is larger in CMIP6 models due to their higher average climate sensitivity. Second, in recent decades, the poleward shift of the NH annual-mean Hadley cell edge is slightly larger in CMIP6 models. Increasing greenhouse gases drive similar trends in CMIP5 and CMIP6 models, so the larger recent NH trends in CMIP6 models point to the role of other forcings, such as aerosols.

## 1 Introduction

The poleward expansion of the Hadley circulation is one of the most robust aspects of the atmospheric general circulation's response to a warming climate in global climate models. This response is seen in models of varying complexity, ranging from idealized aquaplanet simulations (Frierson et al., 2007; Levine and Schneider, 2011; Tandon et al., 2013) to comprehensive general circulation model experiments (Hu et al., 2013; Lu et al., 2007; Tao et al., 2016), such as

those from phases 3 and 5 of the Coupled Model Intercomparison Project (CMIP). The poleward expansion of the Hadley circulation is anticipated to have a number of regional climate impacts in the subtropics, potentially shifting dry regions (Feng and Fu, 2013; Scheff and Frierson, 2012; Schmidt and Grise, 2017), altering zones of ocean upwelling (Cook and Vizy, 2018; Rykaczewski et al., 2015), and modifying hurricane tracks (Kossin et al., 2014; Sharmila and Walsh, 2018; Studholme and Gulev, 2018).

A decade ago, a number of studies began estimating rates of Hadley cell expansion using various observational data

sets (Fu et al., 2006; Hu and Fu, 2007; Seidel and Randel, 2007; Seidel et al., 2008). These rates varied widely by study, ranging from 0.2° to 3° latitude per decade over the period from 1979 until the mid-2000s (Birner et al., 2014; Davis and Rosenlof, 2012; Lucas et al., 2014). The largest observed trends were an order of magnitude larger than those projected by climate models over the same period (Hu et al., 2013; Johanson and Fu, 2009), calling into question whether the observed

trends were biased high and/or whether the models were deficient in simulating circulation trends. Additionally, studies disagreed on the cause of the observed trends. Some studies identified an important role for anthropogenic forcing, including increasing greenhouse gases (Hu et al., 2013; Nguyen et al., 2015; Tao et al., 2016), stratospheric ozone depletion (Kang et al., 2011; McLandress et al., 2011; Min and Son, 2013; Polvani et al., 2011; Son et al., 2010), and changes in anthropogenic aerosols (Allen et al., 2012; Allen and Ajoku, 2016; Kovilakam and Mahajan, 2015). However, other studies

concluded that the observed trends strongly reflected natural climate variability (Allen and Kovilakam, 2017; Amaya et al., 2018; Mantsis et al., 2017).

Recent efforts by the US CLIVAR Working Group on the Changing Width of the Tropical Belt and the International Space Science Institute (ISSI) Tropical Width Diagnostics Intercomparison Project have addressed many of these discrepancies in the previous literature. For example, the large observed rates of expansion documented by some

earlier studies have been attributed to methodological issues. Traditionally, the edge of the Hadley circulation has been defined using the poleward boundary of the zonal-mean meridional mass streamfunction in the mid-troposphere, but departures from mass conservation in reanalyses (particularly in older generation reanalyses) can lead to large spurious trends in the location of the Hadley cell edge defined using the mass streamfunction (Davis and Davis, 2018). Consequently, many studies have sought to estimate trends in the location of the Hadley cell edge using other metrics, including the

transition from zonal-mean surface easterlies to zonal-mean surface westerlies (Grise et al., 2018, hereafter G18; Grise et al., 2019, hereafter G19), the subtropical sea level pressure maximum (Choi et al., 2014), the latitude of the subtropical jet (Maher et al., 2020), the altitude break in tropopause height in the subtropics (Seidel and Randel, 2007; Lucas et al., 2012), thresholds in outgoing longwave radiation (Hu and Fu, 2007; Mantsis et al., 2017), and total column ozone (Hudson et al., 2006). Some of the largest trends in recent decades arise from the metrics derived from tropopause height and outgoing

longwave radiation, but it appears that these metrics are measuring changes unrelated to the poleward expansion of the Hadley circulation. While all of the metrics listed above co-locate climatologically with the poleward boundary of the mass streamfunction, only the surface wind and sea level pressure metrics co-vary interannually with the streamfunction boundary (Davis and Birner, 2017; Davis et al., 2018; Solomon et al., 2016; Waugh et al., 2018), at least in reanalyses and models. Accounting for these issues, estimates of the recent expansion of the Hadley circulation have been narrowed to be $\leq 0.5°$

latitude per decade and within the range of trends indicated by global climate models over the historical period (G18; Staten et al., 2018).

Additionally, in terms of the attribution of the recent trends, G19 concluded that the recent poleward expansion of the Southern Hemisphere (SH) Hadley cell edge was driven in part by anthropogenic forcing (increasing greenhouse gases and stratospheric ozone depletion) and in part by natural variability, whereas the recent poleward expansion of the Northern

Hemisphere (NH) Hadley cell edge was predominantly driven by natural variability. While the observed rates of expansion are approximately comparable in the two hemispheres, models indicate that anthropogenic forcing alone should drive 3–4 times larger expansion in the SH (cf. Fig. 2 of G19). Over the historical period, stratospheric ozone depletion plays a key role in this hemispheric asymmetry, especially during austral summer (DJF). However, even in models forced only by increasing greenhouse gases, the poleward shift of the SH Hadley cell edge is substantially larger than that in the NH (Davis et al., 2016; Grise and Polvani, 2016; Watt-Meyer et al., 2019); only during the SON season are expansion rates comparable between the two hemispheres. G19 concluded that the role of aerosols in the observed Hadley cell expansion appears to be small based on CMIP5 models, but remains very uncertain due to the diverse treatment of aerosols in models.

Most of the conclusions discussed above were formulated using CMIP5 model output, and as CMIP represents an "ensemble of opportunity," it is quite possible that some of the relationships established from CMIP5 models may have been unique to that model generation. The goal of this study is to re-evaluate key conclusions about Hadley cell expansion in a new generation of global climate models (CMIP6) and to assess their robustness across model generation. CMIP6 includes output from updated versions of CMIP5 models (many of which have different treatments of clouds and aerosols, among other factors), as well as new models that did not participate in CMIP5. Overall, we find that the characteristics of Hadley cell expansion are very similar in CMIP5 and CMIP6 models, but we find several notable exceptions, which we detail below.

The paper is organized as follows. Section 2 details the data and methods. Section 3 examines the response of the Hadley cell edge to an idealized 4xCO$_2$ forcing in CMIP6 models and compares the results to CMIP5 models. Section 4 then examines the trends from the historical runs of CMIP6 models, and contrasts them with reanalyses and CMIP5 models. Section 5 briefly compares the 21$^{st}$ century trends in CMIP5 and CMIP6 models. Section 6 provides a summary and concluding thoughts.

## 2 Data and Methods

### 2.1 Data

The primary data used in this study are output from the 24 CMIP5 (Taylor et al., 2012) and 20 CMIP6 (Eyring et al., 2016) models listed in Table 1. These models were selected because they had data available from all of the following runs at the time of the writing of this manuscript:

1) *pre-industrial control*: fully coupled runs simulating 200+ years of unforced variability

2) *historical:* fully coupled runs forced with observed radiative forcings over the period 1850–2005 for CMIP5 and 1850–2014 for CMIP6

3) *AMIP*: atmosphere-only runs forced with observed radiative forcings, sea surface temperatures, and sea ice concentrations over the period 1979–2008 for CMIP5 and 1979–2014 for CMIP6

4) *abrupt 4xCO2*: fully coupled runs in which atmospheric $CO_2$ concentrations are abruptly quadrupled from pre-industrial levels and held fixed for 150 years

Additionally, to examine a high emissions scenario for the 21[st] century, we use the Representative Concentration Pathway (RCP) 8.5 runs (2006–2100) for CMIP5 models and the Shared Socioeconomic Pathway (SSP) 5-8.5 runs (2015–2100) for CMIP6 models. All 24 CMIP5 models have data available for the RCP 8.5 scenario, but only 14 of the 20 CMIP6 models have data available for the SSP 5-8.5 scenario (see CMIP6 models marked with # symbol in Table 1).

For a subset of the models in Table 1, we use three additional runs, which are useful in the attribution of Hadley cell expansion. Following Grise and Polvani (2014), we use the amip4xCO2 and amipFuture (called "amip-future4K" for CMIP6) runs to partition the circulation response to increased atmospheric $CO_2$ into components associated with the direct radiative forcing of $CO_2$ (amip4xCO2 – AMIP) and sea surface temperature (SST) warming (amipFuture – AMIP). The amip4xCO2 runs are atmosphere-only runs with the same SSTs and sea ice as the AMIP runs, but with quadrupled atmospheric $CO_2$ concentrations; the amipFuture runs add a patterned SST anomaly (normalized to a global-mean value of 4K) to the AMIP SSTs, but retain the same $CO_2$ and sea ice concentrations as the AMIP runs (Webb et al., 2017). To determine whether the results are sensitive to the patterned SST anomaly used in the amipFuture runs, we also examine the amip4K (called "amip-p4K" for CMIP6) runs, which add a uniform SST anomaly of 4K to the AMIP SSTs, but retain the same $CO_2$ and sea ice concentrations as the AMIP runs (Webb et al., 2017). 10 CMIP5 models and 7 CMIP6 models have output available for the amip4xCO2, amipFuture, and amip4K runs (see bolded models in Table 1).

Over the historical period (1850–2005 for CMIP5, 1850–2014 for CMIP6), single forcing runs are also examined from available models (see Table S1 for CMIP5 and Table S2 for CMIP6). These runs are identical to the historical runs, except that they only prescribe one forcing over the historical period: well-mixed greenhouse gases, natural (solar and volcanic), anthropogenic aerosols, and ozone. Note that, in CMIP5 models, the ozone-only runs include changes in both stratospheric and tropospheric ozone concentrations, whereas the ozone-only runs in CMIP6 models are only forced by changes in stratospheric ozone concentrations. Furthermore, some CMIP5 models included ozone changes in their greenhouse gas only runs (Gillett et al., 2016), and following G19, we exclude those models here to more clearly separate the influences of stratospheric ozone depletion and increasing greenhouse gases on the circulation response.

To compare the historical circulation trends in models with observations, we make use of the five modern reanalysis data sets listed in Table 2. Because the CFSR reanalysis ends in 2010, we extend it through 2014 using CFSv2. We do not examine the NCEP-NCAR or NCEP-DOE reanalyses here, as they contain substantial departures from mass conservation over the historical period (Davis and Davis, 2018).

## 2.2 Methods

To locate the edges of the Hadley circulation, we make use of two metrics: PSI500 and USFC. PSI500 is defined as the subtropical latitude where the zonal-mean meridional mass streamfunction at 500 hPa switches sign from thermally direct (Hadley circulation) to thermally indirect (Ferrel circulation). USFC is defined as the subtropical latitude where the

zonal-mean zonal wind at the surface switches sign from tropical easterlies to midlatitude westerlies. The metrics are calculated using the Tropical-width Diagnostics code package (TropD; Adam et al., 2018). Before calculating these metrics, the wind fields are zonally and time averaged (i.e., annual-mean, zonal-mean or seasonal-mean, zonal-mean wind fields are used). We note that the NH summertime Hadley circulation is very weak, making it challenging to define the PSI500 metric during some years. We only consider the PSI500 metric from years in which there is a clear crossing of the 500-hPa streamfunction field from positive to negative in the NH subtropics. We consider the PSI500 metric to be undefined if no zero crossing in the streamfunction field occurs or if multiple zero crossings from positive to negative occur within a 20° latitude band ('Lat_Uncertainty = 20' in TropD).

In this paper, we focus on results for the PSI500 metric, as it is the most widely used metric of Hadley cell width in the previous literature. Key results for the USFC metric are shown in the supplementary material. However, when comparing the Hadley cell expansion in models with observations, we show results from both metrics, because of potential biases in the PSI500 metric in reanalyses (Davis and Davis, 2018; G19). We also make brief use of the USFC metric to examine longitudinal asymmetries in the circulation response, as the PSI500 metric can only strictly be defined in the zonal mean. Some recent studies have attempted to generalize the zonal-mean Hadley cell edge (as defined by the PSI500 metric) to individual longitudes by isolating regional meridional overturning cells (Schwendike et al., 2014; Staten et al., 2019). However, interpreting these regional overturning circulations is challenging and remains an area of active research, and thus we do not examine these local overturning cells here.

We evaluate whether the multi-model means of CMIP5 and CMIP6 models are statistically different from one another using a two-tailed Student's t-test. When comparing values from CMIP5 and CMIP6 models, we use large asterisks in the figures to denote where the multi-model means of CMIP5 and CMIP6 models are statistically different at the 95% confidence level. For the significance testing, we treat each model as an independent sample. However, because many climate models are closely related to one another (e.g., Knutti et al., 2013), the actual value of significance is likely to be much lower.

## 3 Dynamical sensitivity of CMIP6 models

Before examining Hadley cell expansion over the historical period, we first compare and contrast the dynamical sensitivity of CMIP5 and CMIP6 models. Following Grise and Polvani (2016, hereafter GP16), we define dynamical sensitivity as the response of the circulation to $4xCO_2$ forcing, which is calculated here as the difference in the Hadley cell edge latitude between its mean position during the last 50 years (years 101-150) of the abrupt $4xCO_2$ run and its mean position in the pre-industrial control run. Examining the dynamical sensitivity is important, as it directly allows us to compare generations of models to a common forcing. The abrupt $4xCO_2$ experiment is chosen for this purpose, as it is a standard experiment planned to be included in all future phases of CMIP (Eyring et al., 2016). In contrast, the forcings used in the historical and future scenario runs of CMIP models change across model generations, making it difficult to verify

whether differences between model generations are because of model improvements or changes in forcings.

Figure 1 shows the response of the NH and SH Hadley cell edge latitudes (as measured by the PSI500 metric) to $4xCO_2$ forcing. Qualitatively similar results for the USFC metric are shown in the supplementary material (Fig. S1). In the SH, both CMIP5 and CMIP6 models show ~2˚ of Hadley cell expansion in response to $4xCO_2$ forcing. The SH expansion has little variation across the seasonal cycle, with slightly larger poleward shifts of the Hadley cell edge in MAM and SON (see also GP16). On average, the poleward expansion seen in CMIP6 models is only slightly larger than that in CMIP5 models, with the difference between CMIP5 and CMIP6 models only being statistically significant in JJA.

In the NH, the response of the Hadley cell edge to $4xCO_2$ has a more dramatic seasonal variation. In the annual mean, the multi-model mean Hadley cell expansion is ~0.75˚ latitude, roughly 40% of the multi-model mean response in the SH. The smaller poleward shift of the NH Hadley cell edge in the annual mean reflects a compensation between a large poleward shift of the NH Hadley cell edge in SON (and to a lesser extent in DJF) and a large equatorward shift of the NH Hadley cell edge in JJA. This seasonality is consistent with previously published results based on CMIP5 models (GP16; Watt-Meyer et al., 2019). The differences between CMIP5 and CMIP6 models are small in all seasons except JJA. However, in JJA, the equatorward contraction of the circulation is notably larger in CMIP6 models. As a result, three CMIP6 models (CESM2, CESM2-WACCM, and SAM0-UNICON) actually contract the annual-mean NH Hadley cell edge equatorward, a result not seen in CMIP5 models (at least as measured by the PSI500 metric).

The differences between CMIP5 and CMIP6 models in Fig. 1 may be because the CMIP6 models, on average, have a higher climate sensitivity (Forster et al., 2019; Zelinka et al., 2020). To check this, in Table 3, we show correlations between the annual-mean global-mean surface temperature response to $4xCO_2$ forcing and the Hadley cell edge response across the inter-model spread of both CMIP5 and CMIP6 models. The results support the conclusions of GP16 based upon CMIP5 models. In the SH, the magnitude of the poleward shift in the Hadley cell edge is strongly correlated with the global-mean surface temperature response throughout the year, with the largest and most significant correlations in MAM and JJA (cf. Fig. 4 of GP16). In other words, models that warm more in response to $4xCO_2$ forcing tend to shift the SH Hadley cell edge further poleward. In contrast, in the NH, the magnitude of the shift in the Hadley cell edge is very poorly correlated with the global-mean surface temperature response in the annual mean. This largely reflects a compensation between a significant positive correlation in DJF and a significant negative correlation in JJA. That is, models that warm more in response to $4xCO_2$ forcing tend to shift the NH Hadley cell edge further poleward in DJF but also further equatorward in JJA. The fact that the only significant differences between CMIP5 and CMIP6 models in Fig. 1 occur in the JJA season in both hemispheres is consistent with Table 3, as JJA is the season with the largest magnitude correlation between the dynamical sensitivity and the global-mean surface temperature response in both hemispheres.

In Fig. 2, we further examine the largest difference between CMIP5 and CMIP6 models identified in Fig. 1: the response of the NH JJA Hadley cell edge to $4xCO_2$ forcing. Figure 2a shows the scatter plot between the responses of the global-mean surface temperature and the NH JJA Hadley cell edge latitude to $4xCO_2$ forcing. As documented in Table 3, the strong anti-correlation between the NH JJA Hadley cell edge shift and the global-mean surface temperature response is

clearly visible. Because CMIP6 models have on average 1 K greater warming in response to 4xCO$_2$ forcing (6.1 K for CMIP6, compared to 5.1 K for CMIP5), the NH JJA Hadley cell edge shifts significantly further equatorward (~4˚ latitude for CMIP6, compared to 1.5˚ latitude for CMIP5).

The timeseries of the response of the NH JJA Hadley cell edge latitude to an abrupt quadrupling of atmospheric CO$_2$ yields further insight into the processes involved (Fig. 2b). Initially, in both CMIP5 and CMIP6 models, the Hadley cell
edge shifts slightly poleward in the first decade after CO$_2$ quadrupling, but then retreats equatorward for the remainder of the 150-year run. Consistent with Figs. 1 and 2a, the equatorward retreat of the NH JJA Hadley cell edge is substantially larger in CMIP6 models.

Following Grise and Polvani (2014) and Shaw and Voigt (2015), we can examine the roles of the direct radiative effects of CO$_2$ and SST warming in this circulation response (see methods in Sect. 2a). In response to a quadrupling of
atmospheric CO$_2$ concentrations (but no change in SSTs), both CMIP5 and CMIP6 models show a ~0.6˚ latitude poleward expansion of the NH JJA Hadley circulation (Fig. 2c), consistent with the immediate circulation response in Fig. 2b after abrupt CO$_2$ quadrupling. In contrast, both CMIP5 and CMIP6 models show a ~1.0˚ latitude equatorward contraction of the NH JJA Hadley circulation in response to a patterned 4K SST warming (with no change in atmospheric CO$_2$ concentrations). NH summer is the season when circulation changes driven by the direct radiative effects of CO$_2$ most clearly oppose those
driven by SST warming (Grise and Polvani, 2014). As argued by Shaw and Voigt (2015), the direct radiative effects of CO$_2$ enhance land-sea temperature contrast and act to shift the circulation poleward, whereas the SST warming reduces land-sea temperature contrast and acts to shift the circulation equatorward. Because the SST warming is larger in CMIP6 models on average (due to their higher climate sensitivity), the SST-driven component of the circulation response would be expected to be larger in CMIP6 models, resulting in a larger net equatorward contraction of the NH Hadley circulation during JJA than
in CMIP5 models. However, as pointed out by Zhou et al. (2019), the exact pattern of SST warming is critical for capturing the equatorward contraction of the NH JJA Hadley cell edge seen in the abrupt 4xCO$_2$ runs. A uniform 4K SST warming would instead result in a poleward expansion of the NH JJA Hadley circulation (Fig. 2c).

One may question the meaningfulness of looking at the NH summertime Hadley circulation, which is generally very weak (Dima and Wallace, 2003) and largely reflects regional overturning circulations in the Indian Ocean/West Pacific
sector (Hoskins et al., 2019). So, to aid in the interpretation of the results in Figs. 1–2, we also examine the regional structure of the NH circulation response during JJA. Figure 3a shows the multi-model mean surface zonal wind response to 4xCO$_2$ forcing for the JJA season for CMIP6 models. From this figure, it is clear that the equatorward contraction of the NH summertime circulation arises largely from the Pacific sector, consistent with findings from CMIP5 models (Shaw and Voigt, 2015; GP16). There is little net shift in the subtropical surface wind field over the Atlantic sector during JJA (see also
Fig. 3c).

The latitude of the transition between tropical surface easterlies and midlatitude surface westerlies over the North Pacific shifts poleward in most seasons but shifts equatorward in summer (Fig. 3b), similar to the zonal-mean Hadley circulation (Fig. 1). In CMIP6 models, the winter and fall circulation shifts further poleward over the Pacific sector than in

the CMIP5 models, but the summer circulation shifts further equatorward. As a result, there is little difference in annual-mean circulation shifts between CMIP5 and CMIP6 models over either the North Pacific or North Atlantic sectors. As noted above for the zonal-mean Hadley circulation (Fig. 2), the equatorward contraction of the Pacific circulation during JJA results from the competing effects of the direct radiative effects of $CO_2$ and SST warming on the circulation (see Fig. S2). The equatorward contraction of the Pacific circulation is larger on average in CMIP6 models (Fig. 3b), as the effect of the warming SSTs overpowers any poleward expansion driven by the direct radiative effects of $CO_2$.

In summary, in this section, we compared and contrasted the responses of the NH and SH Hadley cell edges to abrupt 4x$CO_2$ forcing. The magnitudes and seasonality of the Hadley cell expansion in CMIP6 models are very similar to those in CMIP5 models (Fig. 1). The most notable differences occur in the JJA season, particularly in the NH where CMIP6 models show a substantially larger equatorward contraction of the circulation than CMIP5 models. During this season, the response of the NH Hadley cell edge to 4x$CO_2$ forcing is significantly anti-correlated with the global-mean surface temperature response (Table 3; Fig. 2a), and because the average climate sensitivity of CMIP6 models is larger, the circulation contracts further equatorward in CMIP6 models. This equatorward contraction of the NH Hadley cell during summer largely reflects an equatorward shift of the circulation over the Pacific sector (Fig. 3), where there is a competition between the direct radiative effects of $CO_2$ (which act to expand the circulation poleward) and SST warming (which acts to contract the circulation equatorward). Because the $CO_2$ forcing is the same but the SST warming is larger in CMIP6 models, the net equatorward contraction of the NH summertime circulation is notably larger in CMIP6 models.

## 4 Hadley cell expansion over the historical period

Having compared the models' Hadley cell edge response to a common forcing, we now use this knowledge to compare the models' behavior over the historical period. Figure 4 shows the trends in the annual-mean Hadley cell edge latitude (as measured by both the PSI500 and USFC metrics) over the period 1979–2008 from five reanalyses, CMIP5 models, and CMIP6 models. We examine this 30-year period as it represents the common period covered by the AMIP runs of both CMIP5 and CMIP6 models. Because CMIP5 models' historical runs end in 2005, we have extended these runs with three years of the RCP 8.5 runs until 2008. Qualitatively similar results are found if slightly different end dates are used instead of 2008. For reference, in Fig. 5, we plot the reanalysis and multi-model mean timeseries from which the trends in Fig. 4 are calculated.

Figure 4 shows that the observed trends for the USFC metric (as estimated by reanalyses) are relatively modest ($\leq$ 0.2° latitude per decade in each hemisphere) and within the bounds of the 30-year trends from the control runs of the models (see also G18, G19). In the NH, the reanalysis trends lie at the upper range of trends from the models' historical runs and fall near the multi-model mean trend from the models' AMIP runs, suggesting an important role for SST variability in driving the recent poleward shift in the NH Hadley cell edge (Allen et al., 2014; Allen and Kovilakam, 2017; G19). In the SH, the reanalysis trends compare well with the multi-model mean trends from the historical runs of CMIP5 and CMIP6

models and the multi-model mean trend from the AMIP runs of CMIP5 models. The multi-model mean trend from the AMIP runs of CMIP6 models compares well with the trend from the ERA-5 reanalysis but exceeds the trends from the other reanalyses.

For the PSI500 metric (Fig. 4, left column), trends from the ERA-Interim, MERRA-2, and JRA-55 reanalyses in the NH and from the ERA-Interim reanalysis in the SH are substantially larger than the trends from the models' control runs and greatly exceed the trends from the historical and AMIP runs of most models (see also G18, G19). As discussed by G19, the PSI500 metric is subject to considerable uncertainty in reanalyses (see spread in reanalysis time series in Fig. 5) because of inconsistencies in assimilated satellite radiances across reanalyses (Fujiwara et al., 2017) and departures from mass conservation (Davis and Davis, 2018). By contrast, at least some of the surface pressure and marine surface wind observations are shared among reanalysis centers (Fujiwara et al., 2017), resulting in stronger agreement among the reanalysis time series for the USFC metric (Fig. 5, right column).

Over the 1979–2008 period, the trends from the historical and AMIP runs of CMIP5 and CMIP6 models are very similar, with two key exceptions. First, as noted above, for the USFC metric, the trends in the SH Hadley cell edge are significantly larger in the AMIP runs of CMIP6 models than in the AMIP runs of CMIP5 models (Fig. 4d), but this result is metric dependent and does not hold for the PSI500 metric (Fig. 4c). Second, for both the PSI500 and USFC metrics, the trends in the NH Hadley cell edge are significantly larger in the historical runs of CMIP6 models than in the historical runs of CMIP5 models (Figs. 4a-b). This can also clearly be seen in the time series in Fig. 5 and is not unique to the 1979–2008 period highlighted in Fig. 4. The discrepancy between the historical trends in CMIP5 and CMIP6 models in Fig. 4 is unexpected, as increased $CO_2$ results in very similar trends in the NH annual-mean Hadley cell edge in CMIP5 and CMIP6 models (Fig. 1). Indeed, CMIP6 models forced only with increasing greenhouse gases over the historical period (Fig. 5, orange lines) compare very favorably with the historical runs of CMIP5 models (Fig. 5, solid black lines). This evidence suggests that other forcings (solar/volcanic, aerosol, ozone) could be contributing to the larger NH circulation trends in recent decades in the historical runs of CMIP6 models.

To address the role of different forcings in contributing to trends in the models' historical runs, we examine trends in the Hadley cell edge latitude from all available ensemble members of the historical single forcing runs of CMIP5 and CMIP6 models, updating the results of G19 to include CMIP6 models (see their Fig. 2). Results for the NH Hadley cell edge latitude are shown in Fig. 6, and results for the SH Hadley cell edge latitude are shown in Fig. 7. Recall that these single forcing runs are only available from a small subset of the models (8 CMIP5 models and 9 CMIP6 models, as listed in Tables S1 and S2). Following G19, results are shown for two time periods, 1950–2005 and 1979–2005, where 1950 is the start year of the single forcing runs in some CMIP5 models and 2005 is the end year of the single forcing runs in CMIP5 models.

In the NH, CMIP5 and CMIP6 models agree that increasing greenhouse gases were the dominant forcing contributing to a poleward shift of the annual-mean Hadley cell edge over the second half of the 20[th] century (Fig. 6). However, the poleward trends in the Hadley cell edge latitude in the NH associated with increasing greenhouse gases are ~2–3 times smaller than those in the SH, consistent with the results from the abrupt 4xCO2 runs shown in Fig. 1. The roles

of the remaining forcings (solar/volcanic, aerosol, ozone) are smaller and are of inconsistent sign between CMIP5 and
        CMIP6 models. Natural (solar/volcanic) forcing contributes to a poleward shift of the NH Hadley cell edge over the 1979–
        2005 period in CMIP5 models (Allen et al., 2014), but an equatorward shift of the NH Hadley cell edge over the same period
        in CMIP6 models. Anthropogenic aerosol forcing contributes to a statistically significant equatorward shift of the NH
        Hadley cell edge over the 1950–2005 period in CMIP5 models (Allen and Ajoku, 2016), but this influence has weakened in
CMIP6 models (particularly for the USFC metric). Finally, the ozone single forcing run is associated with a poleward shift
        of the NH Hadley cell edge in CMIP5 models (Allen et al., 2014), but not in CMIP6 models. Here, a large difference
        between CMIP5 and CMIP6 models is expected, as the ozone single forcing runs are driven by both tropospheric and
        stratospheric ozone forcing in CMIP5 models but only by stratospheric ozone forcing in CMIP6 models (which is well
        known to have a much larger effect on the circulation in the SH).

Unfortunately, for this subset of models with single forcing runs, the difference in the historical trends in the NH
        Hadley cell edge latitude between CMIP5 and CMIP6 models (Fig. 6) is smaller than for the entire ensemble of models
        shown in Fig. 4. Consequently, it is difficult to use these runs to fully understand the discrepancies in the models' historical
        runs shown in Figs. 4–5. For the USFC metric, the historical trends from the 9 CMIP6 models with single forcing runs are
        larger than those from the 8 CMIP5 models with single forcing runs (Figs. 6c-6d), consistent with Fig. 4b. Over the 1950–
2005 period, the trends in the historical runs of CMIP5 models reflect a compensation between a poleward shift of the
        Hadley cell edge due to greenhouse gas forcing and an equatorward shift of the Hadley cell edge due to anthropogenic
        aerosol forcing (Fig. 6c). In CMIP6 models, the aerosol influence on the circulation is weaker, allowing the greenhouse gas
        forcing to dominate. A similar but weaker pattern in the trends is seen over the 1979–2005 period for the USFC metric (Fig.
        6d), but not for the PSI500 metric (Fig. 6b). Therefore, while Fig. 6 provides some limited evidence that aerosol forcing
may play a role in the discrepancy in the NH historical circulation trends between CMIP5 and CMIP6 models (Figs. 4–5), it
        is difficult to generalize these conclusions based on a small subset of models to the entire multi-model ensemble. What is
        clear is that the larger historical trends in CMIP6 models over the last several decades appear inconsistent with forcing by
        increasing greenhouse gases alone (compare orange, black, and red lines in Figs. 5a-b).

            Figure 7 shows the trends in the SH Hadley cell edge from the historical single forcing runs of CMIP5 and CMIP6
models for the PSI500 metric for both the annual mean and the DJF season. Results for the USFC metric are shown in Fig.
        S3. The results in Fig. 7 largely support the results from Fig. 2 of G19 based on CMIP5 models alone. Over the second half
        of the 20th century, the models indicate that increasing greenhouse gases and stratospheric ozone depletion (particularly
        during DJF) were the dominant forcings contributing to a poleward shift of the SH Hadley cell edge. There is also some
        suggestion that anthropogenic aerosols contributed to a slight equatorward contraction of the SH Hadley cell edge,
particularly over the 1950–2005 period (see also Choi et al., 2019). The one notable difference in the SH historical trends
        between CMIP5 and CMIP6 models is that the circulation trends associated with the ozone forcing appear to be significantly
        weaker in CMIP6 models. However, only a small number of models conducted the historical ozone forcing runs, and
        unfortunately none of the same modeling centers conducted the runs for both CMIP5 and CMIP6. Therefore, inter-model

differences in the circulation response to ozone forcing likely play a role in the discrepancy between CMIP5 and CMIP6
models seen in Fig. 7, particularly because the magnitude of the austral spring polar lower stratospheric cooling in response
to stratospheric ozone depletion is similar in CMIP5 and CMIP6 models (not shown).  The inclusion of tropospheric ozone
forcing in the CMIP5 single forcing runs may also be a factor.

Finally, we explore the seasonality of the recent trends in the NH and SH Hadley cell edge latitudes.  Time series of
the reanalysis and multi-model mean Hadley cell edge latitudes for all four seasons are shown in Fig. 8.  For reference, we
also plot the 1979–2008 trends from individual reanalyses and models in Fig. S4.  Given the confounding issues with the
PSI500 metric in reanalyses discussed above, we focus on the USFC metric in these figures.

In the NH, the reanalysis time series show near-zero to slightly equatorward trends in the Hadley cell edge during
MAM and JJA and fall close to the multi-model mean of the CMIP historical runs during these seasons (see also G18).
However, during DJF and SON, the reanalysis time series show sizeable (~0.3˚− 0.4˚ latitude per decade) poleward trends in
the Hadley cell edge.  During these seasons, the magnitude of the reanalysis trends is larger than the trends from the
historical and AMIP runs of most models (Fig. S4; see also G18).  In DJF, the AMIP runs of CMIP5 and CMIP6 models
approximate the reanalysis trends better than the historical runs (Fig. 8), suggesting the importance of recent SST variability
in driving the observed NH circulation trends during this season.  In SON, the multi-model mean trends from CMIP6
models' historical and AMIP runs are larger than those from CMIP5 models and are in better agreement with the reanalysis
trends (Fig. 8, compare red and black lines).  Hence, the larger trends in the NH Hadley cell edge in CMIP6 models noted
above in the annual mean most clearly manifest themselves during SON (compare Fig. 5b with Fig. 8).

In the SH, the reanalysis time series show consistent poleward trends in the Hadley cell edge (~0.2˚–0.3˚ latitude
per decade in all seasons but JJA), falling close to the multi-model mean trends from the CMIP5 and CMIP6 historical runs
during all seasons (see also G18).  During all seasons but DJF, the time series of the SH Hadley cell edge from reanalyses
also closely parallels the time series from the CMIP6 runs forced only by increasing greenhouse gases (compare orange and
thick blue lines in the right column of Fig. 8).  However, during DJF, the historical greenhouse gas only runs substantially
underestimate the trends in reanalyses, pointing to the importance of stratospheric ozone depletion in driving SH circulation
trends during this season (Figs. 7c-d), as documented by numerous previous studies (Garfinkel et al., 2015; McLandress et
al., 2011; Min and Son, 2013; Polvani et al., 2011; Son et al., 2010; Waugh et al., 2015).

In summary, in this section, we examined the trends in the latitudes of the NH and SH Hadley cell edges over the
late 20[th] century and early 21[st] century in CMIP5 and CMIP6 models and compared them to trends from five reanalyses.  Our
conclusions largely support the conclusions of recent studies documenting Hadley cell expansion in CMIP5 models (e.g.,
Allen and Kovilakam, 2017; G18; G19).  However, we find that the historical trends in the annual-mean NH Hadley cell
edge latitude are significantly larger over the 1979-2008 period in CMIP6 models (Fig. 4).  One might be tempted to
attribute the larger trends in CMIP6 models to their higher average climate sensitivity, but as shown in Sect. 3, the larger
historical circulation trends in CMIP6 models are actually inconsistent with greenhouse gas forcing, which drives
comparable magnitude shifts in the NH annual-mean Hadley cell edge in CMIP5 and CMIP6 models (Figs. 1 and 6).  We

instead conclude that some other forcing (possibly aerosol forcing, see Fig. 6) must be contributing to the larger historical circulation trends in CMIP6 models.

## 5 Projected Hadley cell expansion over the 21st century

Finally, we briefly compare the 21st century trends from the RCP 8.5 runs of CMIP5 models with those from the SSP 5-8.5 runs of CMIP6 models. Figure 9 shows the time series of the annual-mean NH and SH Hadley cell edge latitudes over the period 1920–2100 based on the PSI500 metric. The time series show the multi-model mean of the historical runs extended through the 21st century with the RCP 8.5 runs for CMIP5 models and the SSP 5-8.5 runs for CMIP6 models. The multi-model mean 20th and 21st century time series for CMIP5 and CMIP6 models are virtually identical. For reference, we provide a scatter plot of the 2015–2100 trends from individual models (as well as the trends by season) in Fig. S5. Given that the RCP 8.5 and SSP 5-8.5 runs are dominated by greenhouse gas forcing, the results in Fig. S5 are very similar to those shown for the 4xCO$_2$ forcing in Fig. 1, but with slightly weaker magnitude.

Following Hawkins and Sutton (2012) and G19, we define a "timescale of emergence" as the time at which the multi-model mean forced circulation response surpasses a given threshold of natural variability (as defined from the models' control runs). For the SH, both the CMIP5 and CMIP6 multi-model mean Hadley cell edge latitudes surpass the one standard deviation threshold of variability in the models' control runs (Fig. 9, gray shading) around the year 2000 (Fig. 9b), suggesting that the circulation response to anthropogenic forcing may have already emerged from natural variability (at least by this measure). This early emergence arises principally from the DJF season (G19; Solomon and Polvani, 2016; Thomas et al., 2015), due in large part to the added influence of stratospheric ozone depletion on the circulation during this season. In this high emissions scenario, the SH annual-mean Hadley cell edge would surpass the two standard deviation threshold of variability in the models' control runs (Fig. 9, gray dashed lines) around the year 2045. This timescale is slightly faster than the timescale of emergence (2060) derived from the Community Earth System Model (CESM) Large Ensemble (G19; Quan et al., 2018).

In the NH, as noted by G19, the circulation response would take much longer to emerge from natural variability. In this high emissions scenario, the NH annual-mean Hadley cell edge would surpass the one standard deviation threshold of variability in the models' control runs between 2060–2070 and would not surpass the two standard deviation threshold of variability in the 21st century (Fig. 9a). Again, this timescale is faster than that noted for the CESM Large Ensemble by G19, who did not find the poleward shift of the NH Hadley cell edge to be large enough to emerge from natural variability in the 21st century in that model. Regardless, the NH circulation response will take much longer to emerge from natural variability than the SH circulation response. This is for two reasons: 1) the larger magnitude response of the Hadley cell edge to increasing greenhouse gases in the SH (Fig. 1) and 2) the slightly larger range of natural variability in the Hadley cell edge latitude in the NH (compare gray shading in Fig. 9a and 9b). Note that, during the SON season, the poleward shift of the NH Hadley cell edge may emerge from natural variability as early as 2040 (not shown), due to the larger NH circulation

response to greenhouse gas forcing during that season (Fig. 1).

## 6 Summary and conclusions

In response to increasing greenhouse gases, global climate models show a robust poleward expansion of the Hadley circulation (Davis et al., 2016; GP16; Watt-Meyer et al., 2019), and numerous lines of observational evidence suggest that the Hadley circulation has already expanded over the last 30–40 years (Birner et al., 2014; Davis and Rosenlof, 2012; Seidel

et al., 2008; Staten et al., 2018). Within the past 5 years, studies have used output from CMIP5 global climate models to better understand the causes of the observed expansion (Allen et al., 2014; Allen and Kovilakam, 2017; G19) and to predict its possible evolution over the 21st century (Hu et al., 2013; Tao et al., 2016). In this paper, we assess whether these conclusions are robust across model generations by examining output from CMIP6 models.

We find strong agreement in the trends in the latitudes of the NH and SH Hadley cell edges from CMIP5 and

CMIP6 models in response to abrupt 4xCO2 (Fig. 1), historical (Fig. 4), and 21st century (Fig. 9) forcings. Specifically, we find a number of features to be robust across model generation:

- Models that warm more in response to $CO_2$ forcing (i.e., models with a higher climate sensitivity) generally shift the SH Hadley cell edge further poleward during all seasons, shift the NH Hadley cell edge further poleward during DJF, but contract the NH Hadley cell edge further equatorward during JJA (Table 3; GP16). The equatorward

contraction of the NH circulation during summer arises from the Pacific sector (Fig. 3; Grise and Polvani, 2014; Shaw and Voigt, 2015).

- In response to $CO_2$ forcing, models shift the annual-mean Hadley cell edge 2–3 times further poleward in the SH than in the NH (Fig. 1; GP16; Watt-Meyer et al., 2019). Only during the SON season is the Hadley circulation expansion comparable in the two hemispheres. This implies that, with continued increases in greenhouse gases, the

circulation response will emerge from natural variability in the 21st century much sooner in the SH than in the NH (Fig. 9; G19).

- Over the last 30–40 years, the magnitude of the Hadley cell expansion indicated by reanalyses using the USFC metric is within the range of trends simulated by CMIP models' historical and AMIP runs (Fig. 4; G19). Large discrepancies between reanalysis and model trends primarily result from examining trends in the PSI500 metric,

which has known biases in reanalyses (Davis and Davis, 2018; G19).

- Observed coupled atmosphere-ocean variability has likely played an important role in recent trends, particularly in the NH (Fig. 4; Allen and Kovilakam, 2017). Increasing greenhouse gases and stratospheric ozone depletion have likely played an important role in recent trends in the SH (Fig. 7).

There are, however, several notable differences in CMIP6 models. First, the equatorward contraction of the NH

summertime circulation is stronger in CMIP6 models, apparently as a result of their higher average climate sensitivity (Fig.

2). Second, over the last 30–40 years, the annual average trends in the NH Hadley cell edge in CMIP6 models' historical runs are slightly larger than those in CMIP5 models' historical runs. This discrepancy is not associated with differences in climate sensitivity, as trends in greenhouse-gas only runs over this time period agree well between CMIP5 and CMIP6 models (Figs. 5–6). The biggest discrepancies in historical circulation trends between CMIP5 and CMIP6 models appear to arise from other forcings (solar/volcanic, anthropogenic aerosol, ozone), which contribute to substantial variance in circulation trends across model generations (Figs. 6–7).

Overall, there is good agreement on the characteristics of Hadley circulation expansion in CMIP5 and CMIP6 models, yet several outstanding issues remain that require further understanding. First, the consistency of the hemispheric and seasonal asymmetries of the circulation trends across model generation attests to their robustness, emphasizing a greater need to better understand the physical mechanisms responsible for these asymmetries (see discussion in Watt-Meyer et al., 2019). Second, a better understanding is needed of the roles of non-greenhouse gas forcings on historical circulation trends and why these trends diverge significantly across model generation. Finally, we have focused almost entirely on zonal-mean circulation trends in this paper. We plan to examine the regional manifestations of these circulation trends in future work.

*Code and data availability*. Code to calculate the PSI500 and USFC metrics is freely available from the TropD package (https://doi.org/10.5281/zenodo.1157043). CMIP5 and CMIP6 model output is freely available from the Lawrence Livermore National Laboratory (https://esgf-node.llnl.gov/search/cmip5/; https://esgf-node.llnl.gov/search/cmip6/). ERA-Interim and ERA-5 reanalysis data are freely available from the European Centre for Medium-Range Weather Forecasts (https://apps.ecmwf.int/datasets/data/interim-full-moda/; https://cds.climate.copernicus.eu/#!/search?text=ERA5&type=dataset). MERRA-2 reanalysis data are freely available from NASA (https://doi.org/10.5067/AP1B0BA5PD2K; https://doi.org/10.5067/2E096JV59PK7). JRA-55, CFSR, and CFSv2 reanalysis data are freely available from the National Center for Atmospheric Research Computational and Information Systems Laboratory Research Data Archive (https://doi.org/10.5065/D60G3H5B; https://doi.org/10.5065/D6DN438J; https://doi.org/10.5065/D69021ZF).

*Author contribution*. KG and SD designed the project, KG performed the formal analysis, and KG and SD prepared the manuscript.

*Competing interests*. The authors declare that they have no conflict of interest.

*Acknowledgements*. We thank Penelope Maher, Karen Rosenlof, and two anonymous reviewers for helpful comments. We acknowledge the World Climate Research Programme, which, through its Working Group on Coupled Modelling, coordinated and promoted CMIP6. We thank the climate modeling groups for producing and making available their model

output, the Earth System Grid Federation (ESGF) for archiving the data and providing access, and the multiple funding

agencies who support CMIP6 and ESGF.

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

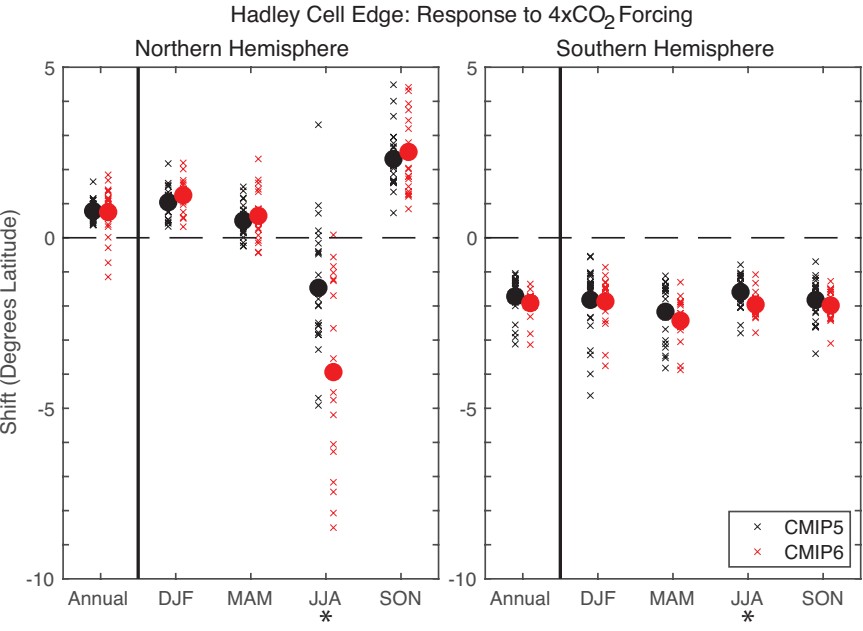

**Figure 1: Response of (left) NH and (right) SH Hadley cell edge latitude (as measured by PSI500 metric) to 4xCO₂ forcing for (black) CMIP5 and (red) CMIP6 models. Here, the response is defined as the difference in the Hadley cell edge latitude between its mean position during the last 50 years (years 101-150) of the abrupt 4xCO₂ run and its mean position in the pre-industrial control run. The response of each model is shown with a small "x", and the multi-model mean response is shown as a large dot. Asterisks denote where multi-model means of CMIP5 and CMIP6 models are statistically different at the 95% confidence level via Student's t-test.**



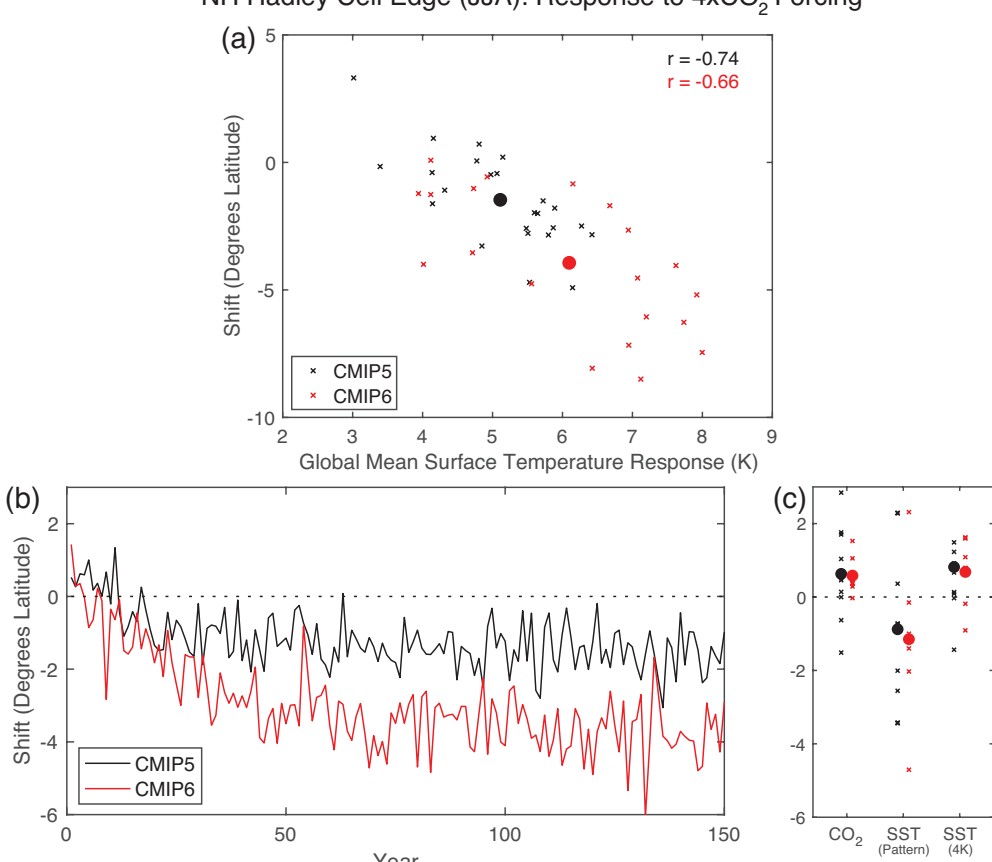

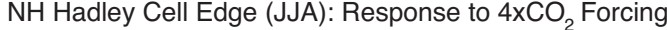

Figure 2: (a) Scatter plot of NH JJA Hadley cell edge response to 4xCO₂ forcing (as measured by PSI500 metric) versus annual-mean global-mean surface temperature response for (black) CMIP5 and (red) CMIP6 models. (b) Time series of NH JJA Hadley cell edge response to abrupt 4xCO₂ forcing for (black) CMIP5 and (red) CMIP6 multi-model mean. (c) Response of NH JJA Hadley cell edge to (first column) quadrupled atmospheric CO₂ concentrations with fixed sea surface temperatures (amip4xCO₂ − AMIP), (second column) patterned sea surface temperature increase with fixed atmospheric CO₂ concentrations (amipFuture − AMIP), and (third column) uniform 4K sea surface temperature increase with fixed atmospheric CO₂ concentrations (amip4K − AMIP).


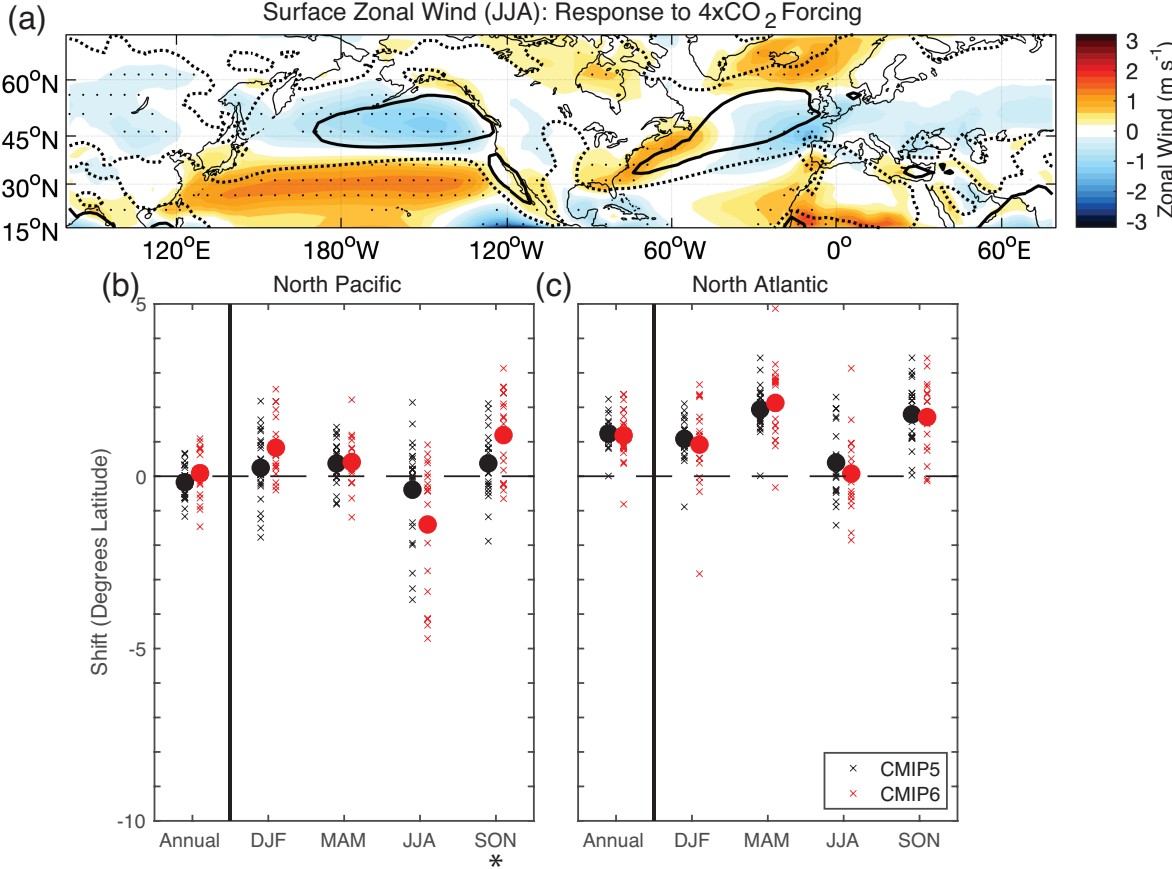


**Figure 3: (a) Multi-model mean response of JJA surface zonal wind field to 4xCO₂ forcing for CMIP6 models. The thick dotted and solid lines indicate the 0 m s⁻¹ and 2.5 m s⁻¹ wind contours from the pre-industrial control climatology, respectively. Stippling indicates where the response is statistically significant at the 95% confidence level via Student's t-test. (b, c) As in Fig. 1, but for**
**the USFC metric calculated over the North Pacific (135°E–125°W) and North Atlantic (60°W–0°E) sectors, respectively.**



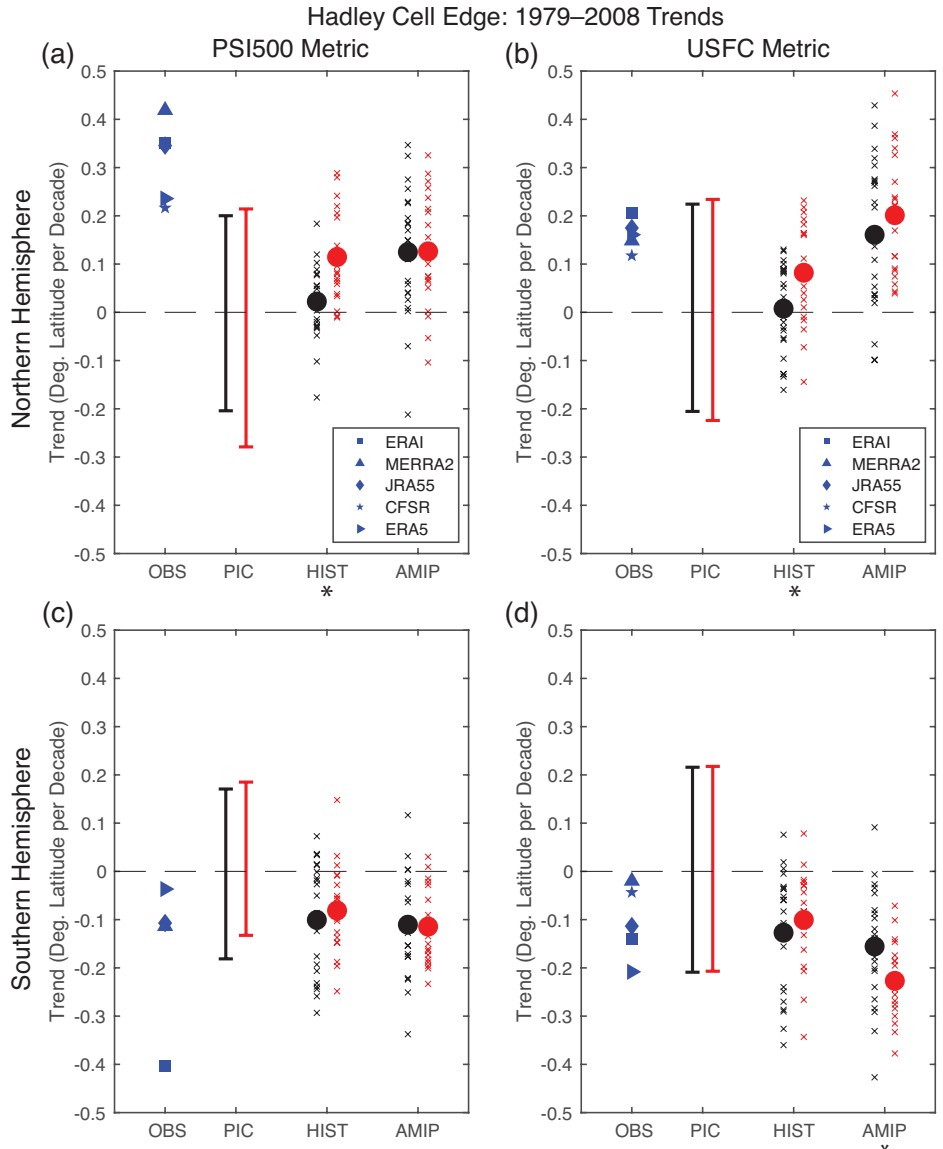

Figure 4: 1979–2008 trends in annual-mean Hadley cell edge latitude, as measured by the (left column) PSI500 and (right column) USFC metrics. Reanalysis trends (OBS, blue symbols) are taken from the ERA-Interim (ERAI), MERRA-2, JRA-55, CFSR, and ERA5 reanalyses. Because MERRA-2 begins in 1980, trends for MERRA-2 are shown for 1980–2008. Control trends (PIC) show the 2.5th–97.5th percentile of trends over 30-year periods from the pre-industrial control runs of (black) CMIP5 and (red) CMIP6 models. Historical trends (HIST) and AMIP trends are calculated from the first ensemble members of (black) CMIP5 and (red) CMIP6 models, where the response of each model is shown with a small "x" and the multi-model mean response is shown as a large dot. Because the historical runs of CMIP5 models end in 2005, they are extended with three years of the RCP 8.5 run until 2008. Asterisks denote where multi-model means of CMIP5 and CMIP6 models are statistically different at the 95% confidence level via Student's t-test.




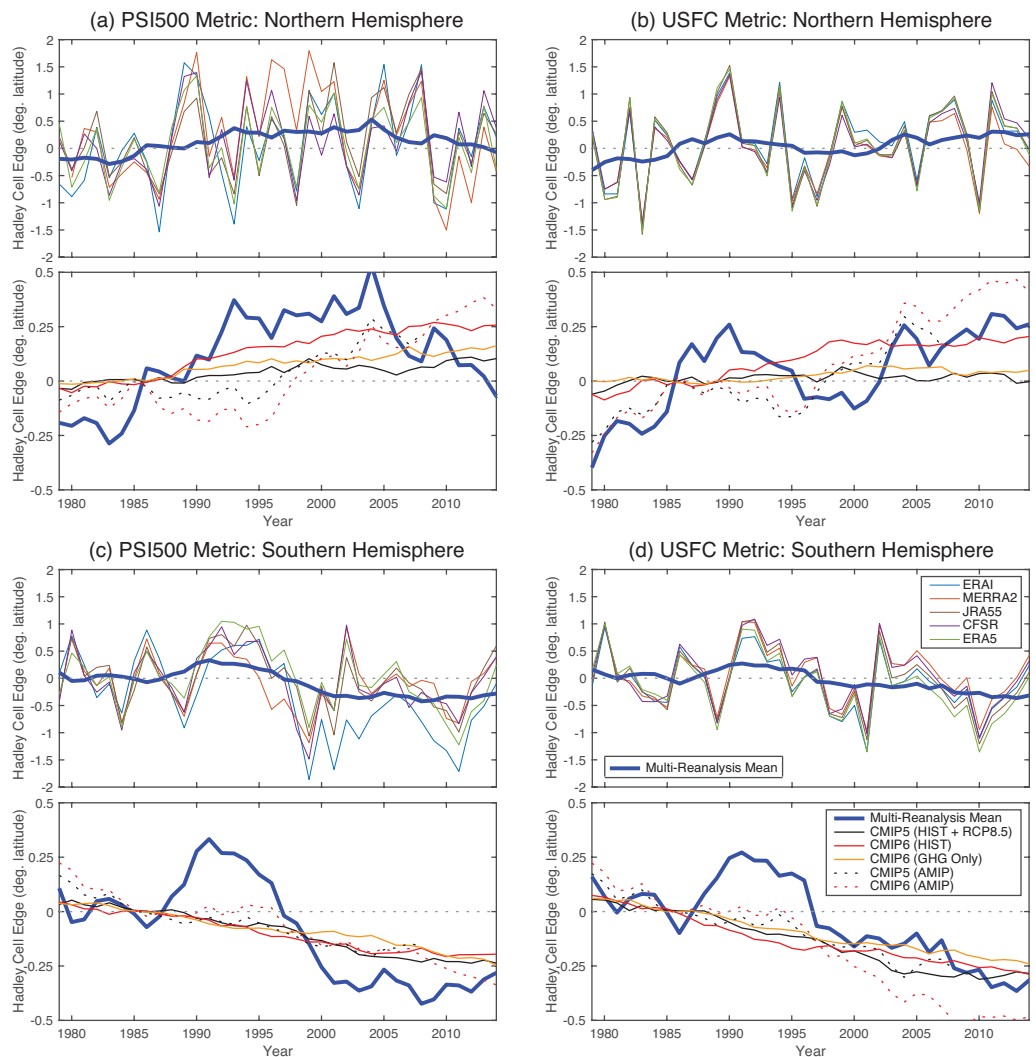

**Figure 5: 1979–2014 time series of annual-mean Hadley cell edge latitude, as measured by the (left column) PSI500 and (right column) USFC metrics. All time series are plotted with respect to their 1980–1990 average. The top set of timeseries in each panel shows five reanalyses (ERA-Interim, MERRA-2, JRA-55, CFSR, and ERA5), along with the multi-reanalysis mean (thick blue line). The bottom set of timeseries in each panel shows the multi-reanalysis mean (thick blue line, reproduced from the plot above), as well as the multi-model mean from (black) CMIP5 historical runs (extended with RCP 8.5 until 2014), (red) CMIP6 historical runs, (orange) CMIP6 historical greenhouse gas-only runs (using all available ensemble members; see Table S2), (black dashed) CMIP5 AMIP runs, and (red dashed) CMIP6 AMIP runs. The reanalysis-mean and multi-model mean time series are smoothed with a ten-year running mean to better visualize the low frequency variability in each time series. Note that the scale is different for the top and bottom sets of timeseries in each panel.**

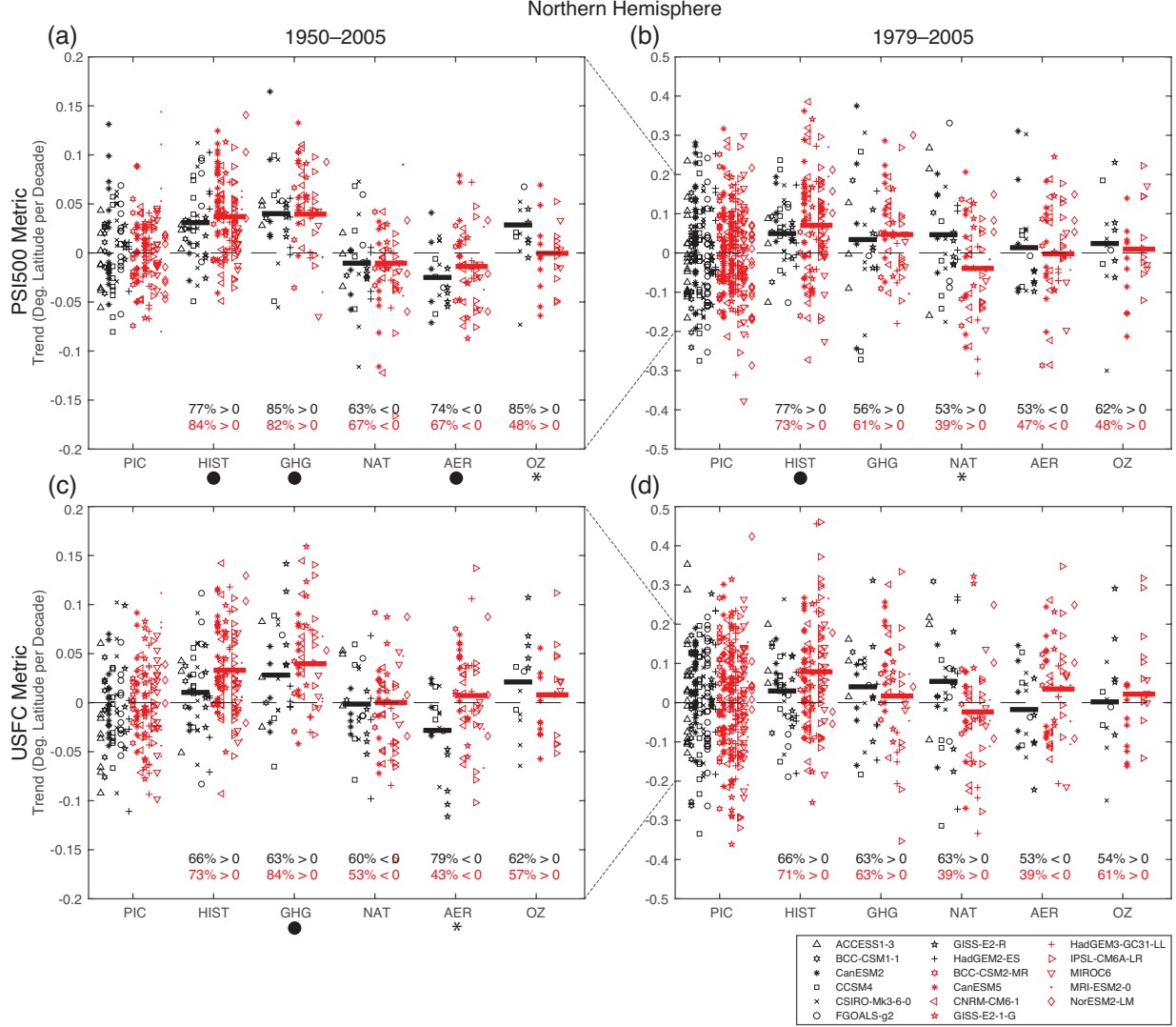

**Figure 6: Trends in annual-mean NH Hadley cell edge latitude over (left column) 1950–2005 and (right column) 1979–2005 for (black) CMIP5 and (red) CMIP6 models. Trends are shown for all available ensemble members of the following runs: (HIST) historical, (GHG) greenhouse gas only, (NAT) solar and volcanic only, (AER) anthropogenic aerosol only, and (OZ) ozone only. Multi-model mean trends are shown as thick horizontal lines. Trends are also shown for all independent time periods of equivalent length from the pre-industrial control (PIC) runs. Large black dots denote forcings with trends statistically different from zero in both CMIP5 and CMIP6 models. Asterisks denote where multi-model means of CMIP5 and CMIP6 models are statistically different at the 95% confidence level via Student's t-test.**

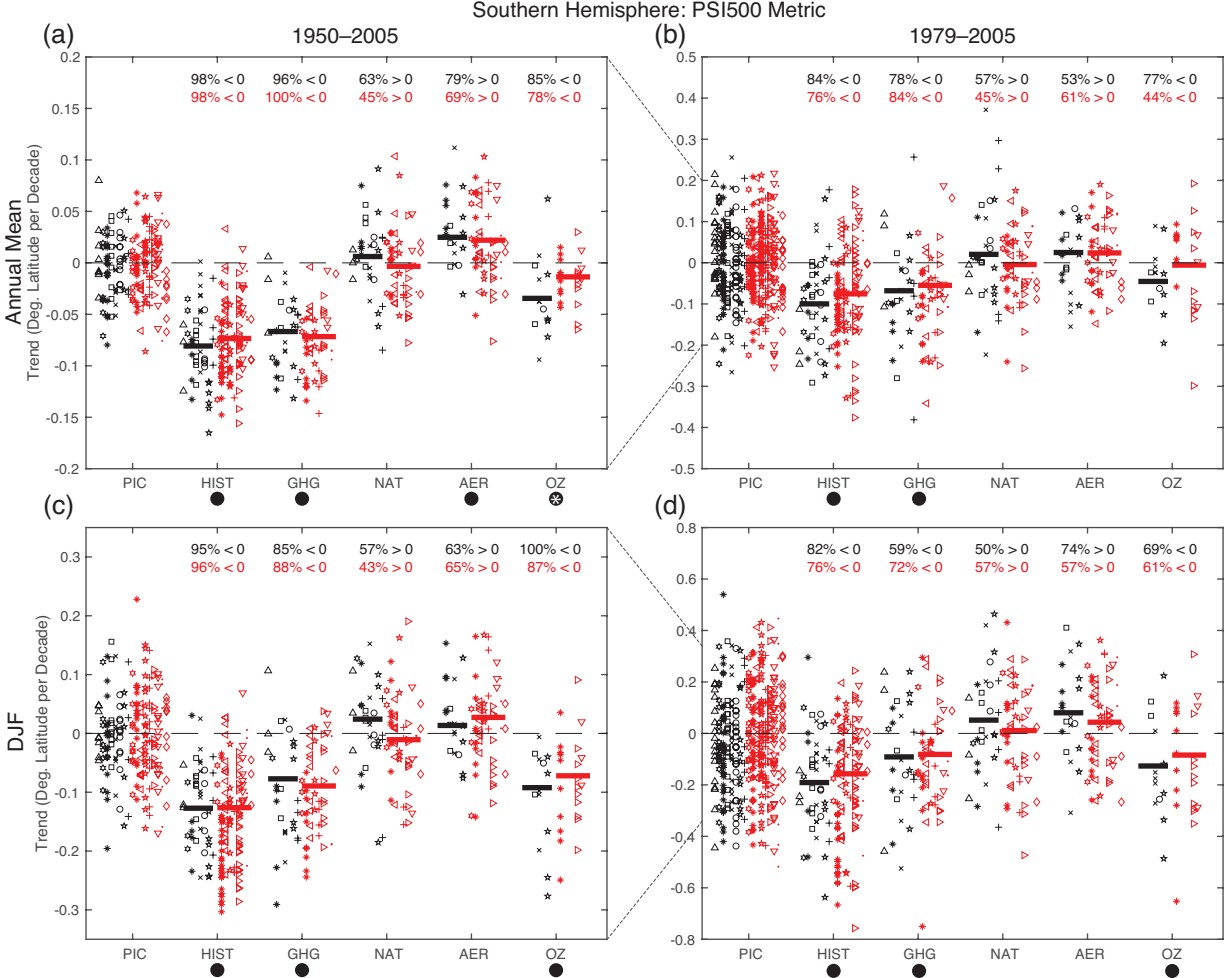

Figure 7: As in Fig. 6, but for (top row) annual-mean SH Hadley cell edge latitude and (bottom row) DJF-mean SH Hadley cell edge latitude (as measured by PSI500 metric).

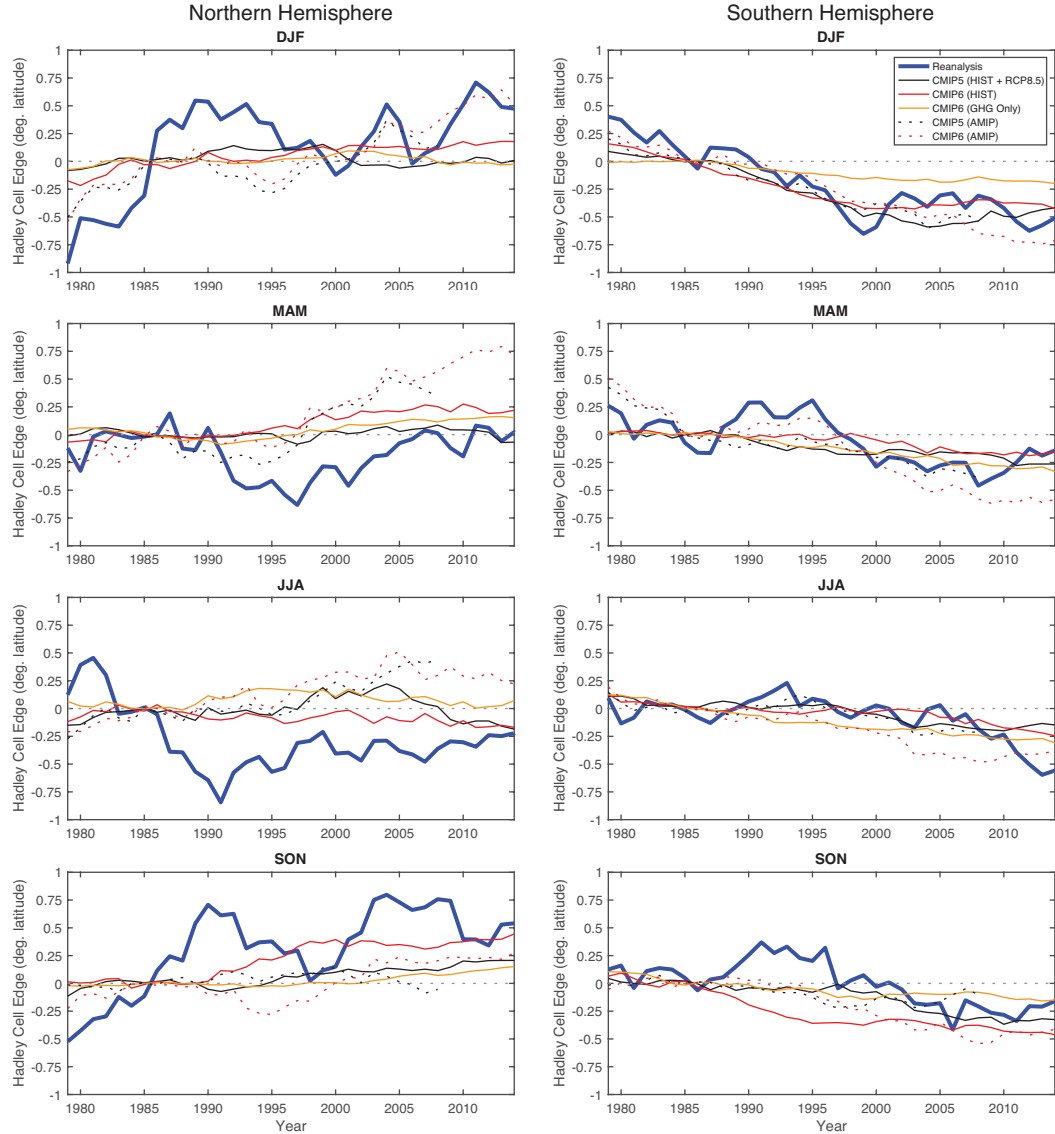

**Figure 8: As in Fig. 5, but for time series of 1979–2014 seasonal-mean Hadley cell edge latitudes (as measured by USFC metric) in**
**both hemispheres.**




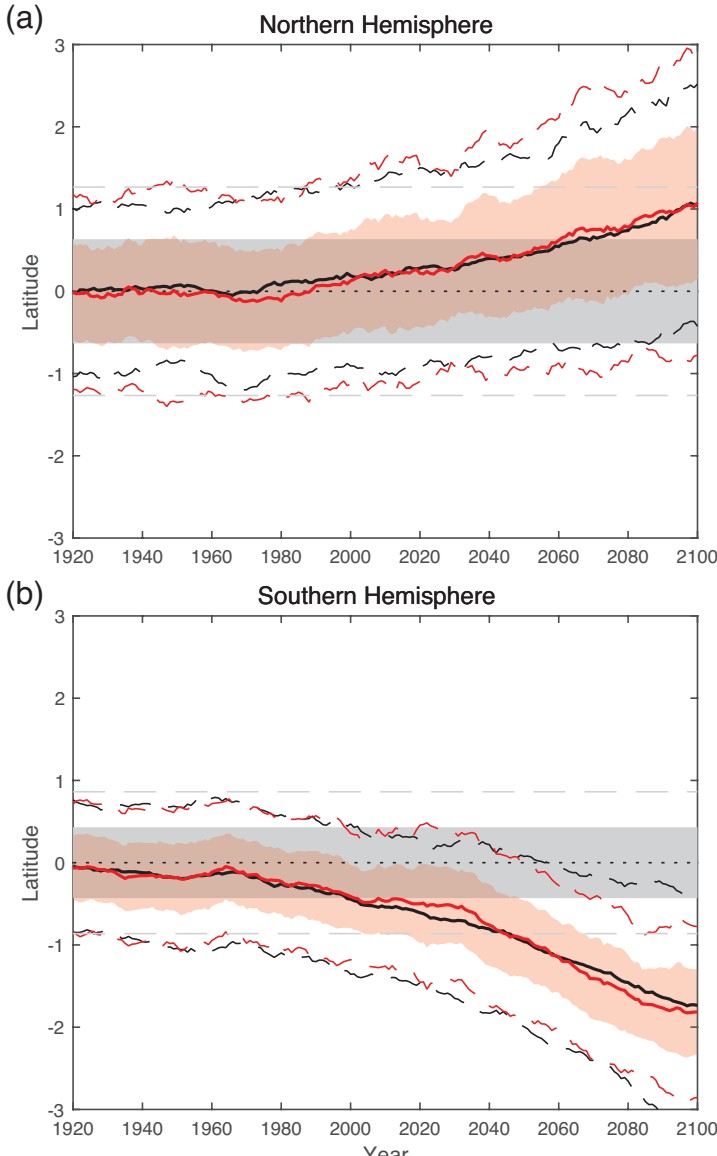

**Figure 9: 1920–2100 time series of annual-mean Hadley cell edge latitude (as measured by PSI500 metric) from CMIP5 historical + RCP 8.5 and CMIP6 historical + SSP 5-8.5 runs. The multi-model mean (smoothed with 10-year running mean) for CMIP5 (CMIP6) models is plotted as a solid black (red) line. One standard deviation range across the inter-model spread of CMIP6**
**models is shown as red shading. Two standard deviation range across the inter-model spread of CMIP5 (CMIP6) models is plotted as black (red) dashed lines. One and two standard deviation ranges about the pre-industrial control latitude of CMIP6 models are shown as gray shading and gray dashed lines, respectively. All latitudes are plotted with respect to the multi-model mean pre-industrial control latitude.**


| CMIP5 Model | Resolution (°lon × °lat) | CMIP6 Model | Resolution (°lon × °lat) |
|---|---|---|---|
| ACCESS1.0 | 1.875° × 1.25° | **BCC-CSM2-MR**[#] | 1.125° × 1.1215° |
| ACCESS1.3 | 1.875° × 1.25° | BCC-ESM1 | 2.8125° × 2.7906° |
| **BCC-CSM1.1** | 2.8125° × 2.7906° | CAMS-CSM1-0[#] | 1.125° × 1.1215° |
| BCC-CSM1.1(m) | 1.125° × 1.1215° | **CanESM5**[#] | 2.8125° × 2.7906° |
| BNU-ESM | 2.8125° × 2.7906° | **CESM2**[#] | 1.25° × 0.9424° |
| **CanESM2 (CanAM4)** | 2.8125° × 2.7906° | CESM2-WACCM[#] | 1.25° × 0.9424° |
| CCSM4 | 1.25° × 0.9424° | **CNRM-CM6-1**[#] | 1.40625° × 1.4008° |
| **CNRM-CM5** | 1.40625° × 1.4008° | CNRM-ESM2-1[#] | 1.40625° × 1.4008° |
| CSIRO Mk3.6.0 | 1.875° × 1.8653° | E3SM-1-0 | 1.0° × 1.0° |
| EC-EARTH | 1.125° × 1.1215° | EC-Earth3[#] | 0.7031° × 0.7018° |
| FGOALS-g2 | 2.8125° × 2.7906° | EC-Earth3-Veg[#] | 0.7031° × 0.7018° |
| GFDL CM3 | 2.5° × 2.0° | GISS-E2-1-G | 2.5° × 2.0° |
| GISS-E2-R | 2.5° × 2.0° | HadGEM3-GC31-LL | 1.875° × 1.25° |
| **HadGEM2-ES (HadGEM2-A)** | 1.875° × 1.25° | **IPSL-CM6A-LR**[#] | 2.5° × 1.2676° |
| INM-CM4.0 | 2.0° × 1.5° | **MIROC6**[#] | 1.40625° × 1.4008° |
| **IPSL-CM5A-LR** | 3.75° × 1.8947° | **MRI-ESM2-0**[#] | 1.125° × 1.1215° |
| IPSL-CM5A-MR | 2.5° × 1.2676° | NESM3[#] | 1.875° × 1.8653° |
| **IPSL-CM5B-LR** | 3.75° × 1.8947° | NorESM2-LM | 2.5° × 1.8947° |
| **MIROC5** | 1.40625° × 1.4008° | SAM0-UNICON | 1.25° × 0.9424° |
| MIROC-ESM | 2.8125° × 2.7906° | UKESM1-0-LL[#] | 1.875° × 1.25° |
| **MPI-ESM-LR** | 1.875° × 1.8653° | | |
| **MPI-ESM-MR** | 1.875° × 1.8653° | | |
| **MRI-CGCM3** | 1.125° × 1.1215° | | |
| NorESM1-M | 2.5° × 1.8947° | | |

**Table 1: Global climate models used in this study. Resolution indicates the horizontal resolution at which the data are provided. Bolded models denote those models with output available from the amip4xCO$_2$, amipFuture/amip-future4K, and amip4K/amip-p4K runs. The # symbols denote the CMIP6 models with output available from the SSP 5-8.5 run. The first ensemble member is used for each model, except in two cases when it is unavailable. In those cases, the 'r8i1p1f1' ensemble member is used for the abrupt 4xCO$_2$ run of EC-Earth3, and the 'r2i1p1f1' ensemble member is used for the amip-p4K run of MIROC6.**


| Reanalysis | Resolution (°lon × °lat) | Time Period | Citation |
|---|---|---|---|
| ERA-5 | 0.25° × 0.25° | 1979–2014 | Hersbach et al. (2019) |
| ERA-Interim | 0.75° × 0.75° | 1979–2014 | Dee et al. (2011) |
| JRA-55 | 1.25° × 1.25° | 1979–2014 | Kobayashi et al. (2015) |
| NASA MERRA-2 | 0.625° × 0.5° | 1980–2014 | Gelaro et al. (2017) |
| NCEP CFSR | 0.5° × 0.5° | 1979–2010 | Saha et al. (2010) |
| CFSv2 | | 2011–2014 | Saha et al. (2014) |

**Table 2:  Reanalysis data sets used in this study.**





|  | Northern Hemisphere | Southern Hemisphere |
|---|---|---|
| Annual | 0.28 | **0.56** |
|  | 0.09 | **0.58** |
| DJF | **0.72** | 0.39 |
|  | **0.55** | **0.51** |
| MAM | 0.15 | **0.55** |
|  | -0.44 | **0.64** |
| JJA | **-0.74** | 0.59 |
|  | **-0.66** | **0.75** |
| SON | 0.40 | **0.44** |
|  | **0.61** | 0.37 |

**Table 3: Correlations between poleward shift of Hadley cell edge latitude in response to 4xCO$_2$ forcing (as measured by PSI500 metric) with annual-mean global-mean surface temperature response to 4xCO$_2$ forcing. Positive correlations imply that models that warm more shift the Hadley cell edge further poleward. Correlations for CMIP5 and CMIP6 models are shown in the top and bottom rows of each cell, respectively. Correlations that are statistically significant at the 95% confidence level via Student's t-test are bolded.**