# Peer review of "Hadley cell expansion in CMIP6 models"

_Atmospheric Chemistry and Physics, 2019_

## Referee Comment (RC1) · Anonymous Referee #1 · 10 Feb 2020

The submission by Davis and Grise sounds at first like turn-the-crank research paper. Indeed, the analysis is a repeat of earlier work by the authors, only with some newer datasets (CMIP6 and ERA5). However, the authors do a commendable job of contrasting their results with their earlier work and under light of other resent studies. In doing so, the authors point out several outstanding questions, making this work a useful step forward.

I have a few mostly-editorial comments.

Line 15-17: the sentence "First, both...,  but this..." would be more clear as "First, while..., this..." to make it clear that "First" does not refer to the first clause, but to the second.

Line 43 and 52: there's another "First" "second" list here, but it's not clearly introduced as a list, and it sounds like the list may continue afterward. "For example" and "In addition" might be better.

Lines 125-128: This is not the Hadley circulation boundary. If you believe the EDJ to be meaningfully related to the HC edge in the different ocean basins, you should state so, and somehow justify your belief.

Line 145: "drastic" is the wrong word; "dramatic" is better

Line 166: Table 1 does not "support" the fact that the only significant differences occur during JJA—that fact doesn't need supporting. But Table 1 helps explain the difference. A more definitive way of explaining this difference would be to normalize the shifts by the respective sensitivities (or remove the component explained by the sensitivity) and determining whether the difference remains significant once the impact of sensitivity is removed.

Line 168: Fig. 2 vs Figure 2a

Line 289: "with forcing" would be clearer as "with a higher sensitivity to"

Line 346: "when which" should be "in which" or "during which"

---

## Referee Comment (RC2) · Penelope Maher (Referee) · 12 Feb 2020

**Review of 'Hadley cell expansion in CMIP6 models'**

*by Kevin M. Grise and Sean M. Davis*

Manuscript number: acp-2019-1206

*Recommendation*: Minor Revision

*Reviewer Name*: Penelope Maher

**1 Summary of the Review**

The manuscript documents contrasts trends in the Hadley cell edge in current generation CMIP6 models, the previous generation CMIP5 models and multiple reanalysis products. This results are new, novel and will be of broad interest to the community. Put simply the manuscript is superb and I very little reservation in recommending this for publication with only a few minor changes. The manuscript is skillfully written and clearly communicated. I particularly liked the perfectly simple and concise title, an excellent and well motivated introduction, and plots of a consistently high quality and common style. The manuscript also has a nice balance of both the NH and SH perspectives.

I would be happy to review the paper a second time should it be neeed.

**2 Minor Comments**

1. In the introduction the focus is very much Hadley cell edge rather than tropical edge. So I was a bit surprised then to see the EDJ metric used in the analysis. I think a more general description of Hadley cell edge vs tropical edge metrics is then needed in the introduction. Then bring this through to your second last paragraph in the introduction that describes the aims of the paper.

2. My interpretation for the use of the EDJ is that your taking advantage of the correlations between the HC edge and the EDJ in order to get longitudinal variability. Can you explain why you used this approach over say a local HC method (eg Schwendike et al. (2014) or your own study Staten et al. (2019))? Is this to avoid adding more metrics (ie consistent with TropD and the idea that fewer/better methods are the goal) or is it that you have concerns about the local methods accuracy/applicability?

3. Introduction: In addition to the papers already cited, there are a number of excellent papers on Hadley cell expansion from Chris Lucas and Hanh Nguyen at the Bureau of Meteorology which could also be cited here. I am thinking papers such as Nguyen et al. (2012, 2015, 2018) Lucas et al. (2012, 2013); Lucas and Nguyen (2015). I am not suggesting you cite them all, but one or two would be good.

4. The introduction would benefit from a brief discussion that the Hadley cell latitude (middle atmosphere mass stream function zero crossing latitude) is thought to be the most reliable measure of tropical expansion (this is touched on briefly in L45 but I think this could be expanded a little). I think there should also be mention to other tropical edge metrics (STJ, EDJ, OLR, P-E etc) which are used within the literature already cited, and include the relationships with the jets (strength and position) in terms of tropical edge metrics and relationships with the HC position Ceppi and Hartmann (2012); Menzel et al. (2019); Maher et al. (2019).

**3 Editorial comments**

**3.1 Introduction**

1. L21: I found 'indicate' awkward in this sentence. I suggest rearranging to the following or similar: ' The poleward expansion of the Hadley circulation is one of the most robust aspects of the atmospheric general circulations response to a warming climate in global circulation models.'

2. The acronyms CMIP, SH and NH are already written in full in the abstract so I do not think you need to define them again in the introduction.

**3.2 Data and Methods**

1. L79-90: I think this would be easier to read directly from Table S1/S2 rather than list in the paragraph. I would then move move the tables from the supplementary into the manuscript. Instead of 'x' you could add the time window of the data, add a column for the reference for each model, add a column for indicating if CMIP5 or 6 (then you only need 1 table) and include the horizontal resolution of the model. This would probably then be a landscape whole page table which is common for CMIP papers. These are simply suggestions, proceed as you wish.

2. L107-112: I would also use a table for the reanalysis data sets.

3. It would be mentioning in this section the method used to test significance (it is stated in each of the plots already).

**3.3 Dynamical sensitivity of CMIP6 models**

1. Fig 1 (and S1): I think this plot would look better and take up less space if it were 1 row and 2 column (ie side by side). I found the asterisk (and in later plots the circles) hard to interpret (are they in S1 too?). I think open and filled circles for the ensemble mean might communicate this clearer or have a lighter and darker versions of black/red. Likewise for Fig 4

2. Was the goal of Fig 2-3 on focusing on the NH only to draw out the differences seen in Fig 1 for JJA in NH?

**3.4 Hadley cell expansion over the historical period**

1. Fig 5: The legends are a little small and repeated. Suggest making larger and only having one (perhaps at the top or bottom of the panel). Might also help to add 'ensemble mean' for the purple line in the top panels of each sub-plot.

2. L238: only 3 out of the models 'greatly exceed' historical and AMIP runs. Suggest mentioning the reanalysis models by name or putting in the clause 'some of the reanalysis greatly exceed'.

3. Fig 6-7: I can't really tell one model from another. I am not sure this level of detail is helpful as I found these plots a little overwhelming. The ensemble mean bars are also easily lost in the scatter. This is my personal opinion and the authors can change or not change these figures as they wish.

4. What is happening in Fig c-d bottom panels for the reanalysis between 1990-2000 – is this the PDO?

**3.5 21st century trends**

1. Suggest starting this section with 'The' so that the 5 and 21 are separated.

**3.6 Supplementary**

1. The first page would benefit from adding a title of the paper and stating it is the supplementary material (minimum) or adding title page (if you wish).

**References**

Ceppi, P. and Hartmann, D. L. (2012). On the speed of the eddy-driven jet and the width of the hadley cell in the southern hemisphere. *J. Climate*, 26(10):3450–3465.

Lucas, C. and Nguyen, H. (2015). Regional characteristics of tropical expansion and the role of climate variability. *Journal of Geophysical Research: Atmospheres*, 120(14):6809–6824.

Lucas, C., Nguyen, H., and Timbal, B. (2012). An observational analysis of southern hemisphere tropical expansion. *Journal of Geophysical Research: Atmospheres*, 117(D17):n/a–n/a. D17112.

Lucas, C., Timbal, B., and Nguyen, H. (2013). The expanding tropics: a critical assessment of the observational and modeling studies. *WIREs Clim Change*.

Maher, P., Kelleher, M., Sansom, P., and Methven, J. (2019). Is the subtropical jet shifting poleward? *Climate Dynamics*. 10.1007/s00382-019-05084-6.

Menzel, M. E., Waugh, D., and Grise, K. (2019). Disconnect Between Hadley Cell and Subtropical Jet Variability and Response to Increased CO2. *Geophysical Research Letters*, 46(12):7045–7053.

Nguyen, H., Evans, A., Lucas, C., Smith, I., and Timbal, B. (2012). The hadley circulation in reanalyses: Climatology, variability, and change. *J. Climate*, 26(10):3357–3376.

Nguyen, H., Hendon, H. H., Lim, E. P., Boschat, G., Maloney, E., and Timbal, B. (2018). Variability of the extent of the hadley circulation in the southern hemisphere: a regional perspective. *Climate Dynamics*, 50(1):129–142.

Nguyen, H., Lucas, C., Evans, A., Timbal, B., and Hanson, L. (2015). Expansion of the southern hemisphere hadley cell in response to greenhouse gas forcing. *J. Climate*, 28(20):8067–8077.

Schwendike, J., Govekar, P., Reeder, M. J., Wardle, R., Berry, G. J., and Jakob, C. (2014). Local partitioning of the overturning circulation in the tropics and the connection to the hadley and walker circulations. *Journal of Geophysical Research: Atmospheres*, 119(3):1322–1339.

Staten, P. W., Grise, K. M., Davis, S. M., Karnauskas, K., and Davis, N. (2019). Regional widening of tropical overturning: Forced change, natural variability, and recent trends. *Journal of Geophysical Research: Atmospheres*, 124(12):6104–6119.

---

## Referee Comment (RC4) · Anonymous Referee #3 · 22 Feb 2020

General comments:

This is a very thorough and thoughtful study on the expansion of the Hadley cell across CMIP5 models/CMIP6 models/reanalyses. It clearly lays out the key similarities and differences between CMIP5 and CMIP6, and sets the differences between the models and reanalysis in a useful context. I have one main comment, and if it is addressed, the manuscript would be suitable for publication.

My comment is that the role of the pattern of SST warming for the NH JJA contraction should be discussed. For CMIP5 models, there are significant differences in Hadley cell edge response between the amipFuture and amip4K experiments for the NH JJA season. So it would be helpful if results from the amip4K experiment were shown for comparison. Furthermore, Zhou et al. (2019) argue that an ITCZ shift related to the

pattern of enhanced equatorial SST warming drives the subtropical circulation contraction during NH JJA. This is different from the present manuscript, in which it is argued that a general SST warming (pattern not required) forces the equatorial contraction during summer.

Minor comments: 1) How do you define the PSI500 Hadley cell edge for NH JJA if PSI never becomes positive in the tropics, and hence a zero-crossing does not exist? In my experience this occurs for some years in certain models. 2) How exactly is the "response" defined for the abrupt4xCO2 experiment (Fig. 1)? I.e. over what period of the abrupt4xCO2 experiment are you averaging?

References:

Zhou, W., S.-P. Xie and D. Yang (2019): Enhanced equatorial warming causes deep-tropical contraction and subtropical monsoon shift. Nature Climate Change. 9. 834-839.

---

## Author Response (AR1)

**Response to reviewer #1's comments on Grise and Davis (2020)**

We would like to thank the reviewer for taking time to review our manuscript and providing helpful comments. Based on the reviewer's comments, we have made a number of minor changes and clarifications to the manuscript. Detailed point-by-point responses to all comments are provided below, and the original reviewer's comments are provided in bold type.

**The submission by Davis and Grise sounds at first like turn-the-crank research paper. Indeed, the analysis is a repeat of earlier work by the authors, only with some newer datasets (CMIP6 and ERA5). However, the authors do a commendable job of contrasting their results with their earlier work and under light of other recent studies. In doing so, the authors point out several outstanding questions, making this work a useful step forward. I have a few mostly-editorial comments.**

We thank the reviewer for this recognition of our efforts to characterize, discuss, and interpret the CMIP6 results in light of previous findings.

**Line 15-17: the sentence "First, both. . ., but this. . ." would be more clear as "First, while. . ., this. . ." to make it clear that "First" does not refer to the first clause, but to the second.**

Thanks. Following the reviewer's suggestion, we have corrected this sentence to the following on lines 15–17 in the revised manuscript:

"First, while both CMIP5 and CMIP6 models contract the NH summertime Hadley circulation equatorward (particularly over the Pacific sector), this contraction is larger in CMIP6 models due to their higher average climate sensitivity."

**Line 43 and 52: there's another "First" "second" list here, but it's not clearly introduced as a list, and it sounds like the list may continue afterward. "For example" and "In addition" might be better.**

Thanks. We have changed "first" to "for example" (line 44) and "second" to "additionally" (line 62).

**Lines 125-128: This is not the Hadley circulation boundary. If you believe the EDJ to be meaningfully related to the HC edge in the different ocean basins, you should state so, and somehow justify your belief.**

Given the comments from Reviewers #1 and #2, we have eliminated the usage of the EDJ metric from the manuscript, as it does not directly correspond to the location of the Hadley cell edge (although it is highly correlated with it in terms of

interannual variability and its response to climate change; see Waugh et al. 2018). We now use the USFC metric to quantify longitudinal variability in the tropical edge. The results are very similar over the Pacific sector, which is the focus of our discussion (see new Fig. 3 and Fig. S2).

**Line 145: "drastic" is the wrong word; "dramatic" is better**

Thanks. We have changed "drastic" to "dramatic" on line 168 in the revised manuscript.

**Line 166: Table 1 does not "support" the fact that the only significant differences occur during JJA. That fact doesn't need supporting. But Table 1 helps explain the difference.**

We have changed "is supported by" to "is consistent with" on line 189 in the revised manuscript.

**A more definitive way of explaining this difference would be to normalize the shifts by the respective sensitivities (or remove the component explained by the sensitivity) and determining whether the difference remains significant once the impact of sensitivity is removed.**

In the SH, where the JJA Hadley cell edge shifts are all of the same sign and of a similar order of magnitude, the reviewer's suggestion works well. If the SH JJA Hadley cell edge shifts are divided by the global-mean surface temperature increase in each model, the difference between CMIP5 and CMIP6 models is no longer statistically significant, supporting our argument in the text and confirming the reviewer's idea.

In the NH, the JJA Hadley cell edge shifts are of varying sign (poleward and equatorward) and of varying orders of magnitude (near-zero to almost 10˚), so applying a simple normalization procedure is not straightforward to interpret (i.e., you are dividing large negative Hadley cell shifts by large positive climate sensitivities but you are dividing small negative, near-zero, and large positive Hadley cell shifts by smaller positive climate sensitivities). If we confine our analysis to only those models with equatorward Hadley cell edge shifts greater than 0.5˚ latitude, if the NH JJA Hadley cell edge shifts in these models are divided by the global-mean surface temperature increase, the difference between CMIP5 and CMIP6 models is no longer statistically significant. However, this result breaks down once models with near-zero and poleward Hadley cell edge shifts are included.

We thank the reviewer for the suggestion of normalizing the Hadley cell shifts by global-mean surface temperature, but because it is not straightforward to apply in the NH, we choose not to include these results in the paper.

As the reviewer also suggests, one can also use linear regression analysis to remove the variance in the Hadley cell edge shifts associated with the variance in climate sensitivity across models, but this analysis can only be used to remove the variance associated with the climate sensitivity. It does not provide any information about the contribution of the climate sensitivity to the mean Hadley cell edge shift in CMIP5 and CMIP6 models, and thus it cannot be used to assess whether the mean Hadley cell edge shifts in CMIP5 and CMIP6 models are related to the difference in mean climate sensitivity.

One can, however, compare the linear regression fits between the global-mean surface temperature response and the Hadley cell edge shifts in CMIP5 and CMIP6 models. The linear regression lines have very similar slopes in both NH JJA (approximately -1.5˚ latitude/Kelvin for CMIP5 and -1.25˚ latitude/Kelvin for CMIP6) and SH JJA (approximately -0.3˚ latitude/Kelvin and -0.2˚ latitude/Kelvin from CMIP6), suggesting that the greater mean climate sensitivity in CMIP6 models contributes to the greater dynamical sensitivity during JJA in both hemispheres. This can clearly be seen in Fig. 2a for NH JJA, as the scatter of points from both CMIP5 and CMIP6 models generally falls along the same diagonal line from the upper left toward lower right.

**Line 168: Fig. 2 vs Figure 2a**

We don't understand the reviewer's comment here. The first sentence introduces Fig. 2 as a whole, whereas the second sentence discusses specifics only in panel a of Fig. 2. We believe that the text is correct as written. Additionally, per ACP guidelines, the abbreviation "Fig." is used when a figure is referenced within a sentence, whereas the word "Figure" is spelled out at the beginning of a sentence.

**Line 289: "with forcing" would be clearer as "with a higher sensitivity to"**

We apologize that our initial wording was confusing. We are actually not discussing the relationship with climate sensitivity here, but are instead referring to the difference between the greenhouse-gas only runs and the full historical runs. We have added a parenthetical note "(compare orange, black, and red lines in Figs. 5a-b)" to clarify this to the reader on line 318 in the revised manuscript.

**Line 346: "when which" should be "in which" or "during which"**

We have changed "when which" to "at which" on line 374 of the revised manuscript.

**Response to reviewer #2's comments on Grise and Davis (2020)**

We would like to thank the reviewer for taking time to review our manuscript and providing helpful comments. Based on the reviewer's comments, we have made a number of minor changes and clarifications to the manuscript. Detailed point-by-point responses to all comments are provided below, and the original reviewer's comments are provided in bold type.

**The manuscript documents and contrasts trends in the Hadley cell edge in current generation CMIP6 models, the previous generation CMIP5 models and multiple reanalysis products. The results are new, novel and will be of broad interest to the community. Put simply the manuscript is superb and I very little reservation in recommending this for publication with only a few minor changes. The manuscript is skillfully written and clearly communicated. I particularly liked the perfectly simple and concise title, an excellent and well-motivated introduction, and plots of a consistently high quality and common style. The manuscript also has a nice balance of both the NH and SH perspectives. I would be happy to review the paper a second time should it be needed.**

We thank the reviewer for her very positive assessment of our manuscript.

**In the introduction the focus is very much Hadley cell edge rather than tropical edge. So I was a bit surprised then to see the EDJ metric used in the analysis. I think a more general description of Hadley cell edge vs tropical edge metrics is then needed in the introduction. Then bring this through to your second last paragraph in the introduction that describes the aims of the paper.**

As discussed below, we have eliminated the usage of the EDJ metric in the analysis. As such, we believe that the aims of the paper in the Introduction are correct as stated.

**My interpretation for the use of the EDJ is that your taking advantage of the correlations between the HC edge and the EDJ in order to get longitudinal variability. Can you explain why you used this approach over say a local HC method (eg Schwendike et al. (2014) or your own study Staten et al. (2019))? Is this to avoid adding more metrics (ie consistent with TropD and the idea that fewer/better methods are the goal) or is it that you have concerns about the local methods accuracy/applicability?**

Given the comments from Reviewers #1 and #2, we have eliminated the usage of the EDJ metric from the manuscript, as it does not directly correspond to the location of the Hadley cell edge (although it is highly correlated with it). We now use the USFC metric to quantify longitudinal variability in the tropical edge. We appreciate the reviewer's recognition of our previous

work that attempts to define the Hadley cell at individual longitudes through local overturning circulations. However, this concept is still relatively new and remains an area of active research, so we feel that using the USFC metric at individual longitudes will be more straightforward for readers to interpret.

To address the reviewer's concern, we have added the following text into the methods section on lines 140-145 of the revised manuscript:

"We also make brief use of the USFC metric to examine longitudinal asymmetries in the circulation response, as the PSI500 metric can only strictly be defined in the zonal mean. Some recent studies have attempted to generalize the zonal-mean Hadley cell edge (as defined by the PSI500 metric) to individual longitudes by isolating regional meridional overturning cells (Schwendike et al., 2014; Staten et al., 2019). However, interpreting these regional overturning circulations is challenging and remains an area of active research, and thus we do not examine these local overturning cells here."

**Introduction: In addition to the papers already cited, there are a number of excellent papers on Hadley cell expansion from Chris Lucas and Hanh Nguyen at the Bureau of Meteorology which could also be cited here. I am thinking papers such as Nguyen et al. (2012, 2015, 2018) Lucas et al. (2012, 2013); Lucas and Nguyen (2015). I am not suggesting you cite them all, but one or two would be good.**

We have added citations to Lucas et al. (2012), Lucas et al. (2014), and Nguyen et al. (2015) in the introduction (lines 33, 37, 52). Many of their other papers focus on the tropopause height metric of tropical width, which does not co-vary interannually with the Hadley cell edge (e.g., Waugh et al. 2018), and/or regional aspects of tropical widening, which are not the focus of the text in the introduction.

**The introduction would benefit from a brief discussion that the Hadley cell latitude (middle atmosphere mass stream function zero crossing latitude) is thought to be the most reliable measure of tropical expansion (this is touched on briefly in L45 but I think this could be expanded a little). I think there should also be mention to other tropical edge metrics (STJ, EDJ, OLR, P-E etc) which are used within the literature already cited, and include the relationships with the jets (strength and position) in terms of tropical edge metrics and relationships with the HC position Ceppi and Hartmann (2012); Menzel et al. (2019); Maher et al. (2019).**

We have added the following text into the introduction to more fully describe the metrics used to define the edges of the tropics (lines 45–58 of the revised manuscript):

"Traditionally, the edge of the Hadley circulation has been defined using the poleward boundary of the zonal-mean meridional mass streamfunction in the mid-troposphere, but departures from mass conservation in reanalyses (particularly in older

generation reanalyses) can lead to large spurious trends in the location of the Hadley cell edge defined using the mass streamfunction (Davis and Davis, 2018). Consequently, many studies have sought to estimate trends in the location of the Hadley cell edge using other metrics, including the transition from zonal-mean surface easterlies to zonal-mean surface westerlies (Grise et al., 2018, hereafter G18; Grise et al., 2019, hereafter G19), the subtropical sea level pressure maximum (Choi et al., 2014), the latitude of the subtropical jet (Maher et al., 2020), the altitude break in tropopause height in the subtropics (Seidel and Randel, 2007; Lucas et al., 2012), thresholds in outgoing longwave radiation (Hu and Fu, 2007; Mantsis et al., 2017), and total column ozone (Hudson et al., 2006). Some of the largest trends in recent decades arise from the metrics derived from tropopause height and outgoing longwave radiation, but it appears that these metrics are measuring changes unrelated to the poleward expansion of the Hadley circulation. While all of the metrics listed above co-locate climatologically with the poleward boundary of the mass streamfunction, only the surface wind and sea level pressure metrics co-vary interannually with the streamfunction boundary (Davis and Birner, 2017; Davis et al., 2018; Solomon et al., 2016; Waugh et al., 2018), at least in reanalyses and models. ”

We do not feel that it is necessary to provide a detailed discussion of the relationships among the strengths and positions of the subtropical and eddy-driven jets, particularly because we no longer use the EDJ metric in the manuscript. The focus of the manuscript is on the Hadley cell edge, not on the jets.

**L21: I found 'indicate' awkward in this sentence. I suggest rearranging to the following or similar: 'The poleward expansion of the Hadley circulation is one of the most robust aspects of the atmospheric general circulations response to a warming climate in global circulation models.'**

Following the reviewer's suggestion, we have changed the first sentence of the introduction (lines 21-22 of the revised manuscript) to the following:

"The poleward expansion of the Hadley circulation is one of the most robust aspects of the atmospheric general circulation's response to a warming climate in global climate models."

**The acronyms CMIP, SH and NH are already written in full in the abstract so I do not think you need to define them again in the introduction.**

According to the ACP manuscript preparation guidelines, abbreviations "need to be defined in the abstract and then again at the first instance in the rest of the text." So, per the guidelines, we define the acronyms in both the abstract and main text.

**L79-90: I think this would be easier to read directly from Table S1/S2 rather than list in the paragraph. I would then**

**move the tables from the supplementary into the manuscript. Instead of 'x' you could add the time window of the data, add a column for the reference for each model, add a column for indicating if CMIP5 or 6 (then you only need 1 table) and include the horizontal resolution of the model. This would probably then be a landscape whole page table which is common for CMIP papers.  These are simply suggestions, proceed as you wish.**

Following the reviewer's suggestion, we have removed Tables S1 and S2 and placed the salient information into a new table (Table 1) in the main text.  We have also added the horizontal resolution of each model into the table, as the reviewer requested. However, we do not include the references for each of the 44 individual models used in this study, as we found it difficult to ascertain the appropriate references for some of the models.  Referring to the citation requirements stated on the CMIP website (https://pcmdi.llnl.gov/CMIP6/TermsOfUse/TermsOfUse6-1.html), we follow their recommendation to cite the relevant articles published in the CMIP6 special issue of GMD.  No mention is made of citing the papers from the individual modeling centers.

To keep Table 1 relatively concise, we have chosen to retain the information about the time periods of the runs in the first paragraph of section 2.1.  However, we have rewritten this paragraph to include an enumerated list of the different model runs (lines 91-97), which will hopefully make this paragraph easier to read.

**L107-112: I would also use a table for the reanalysis data sets.**

The details of the reanalysis data sets are now listed in Table 2.

**It would be mentioning in this section the method used to test significance (it is stated in each of the plots already).**

We had added the following paragraph at the end of section 2b to address how we calculate statistical significance (lines 146-151 of the revised manuscript):

"We evaluate whether the multi-model means of CMIP5 and CMIP6 models are statistically different from one another using a two-tailed Student's t-test.  When comparing values from CMIP5 and CMIP6 models, we use large asterisks in the figures to denote where the multi-model means of CMIP5 and CMIP6 models are statistically different at the 95% confidence level. For the significance testing, we treat each model as an independent sample.  However, because many climate models are closely related to one another (e.g., Knutti et al., 2013), the actual value of significance is likely to be much lower."

**Fig 1 (and S1): I think this plot would look better and take up less space if it were 1 row and 2 column (ie side by side).**

Following the reviewer's suggestion, we have reformatted Fig. 1, Fig. S1, and Fig. S5 in a 1 row-2 column format.

**I found the asterisk (and in later plots the circles) hard to interpret (are they in S1 too?). I think open and filled circles for the ensemble mean might communicate this clearer or have a lighter and darker versions of black/red. Likewise for Fig 4**

We have added the following text into section 2b (lines 147-149 of the revised manuscript) to clarify the meaning of the asterisks in the figures:

"When comparing values from CMIP5 and CMIP6 models, we use large asterisks in the figures to denote where the multi-model means of CMIP5 and CMIP6 models are statistically different at the 95% confidence level."

We appreciate the reviewer's suggestion, but this would not be straightforward to apply in Figures 6–7, where two different significant tests are applied (i.e., testing whether the mean trend in both CMIP5 and CMIP6 models is statistically different from zero and testing whether the multi-model means of CMIP5 and CMIP6 models are statistically different). We prefer to use consistent symbols and formatting across all figures, so we prefer to retain the use of the asterisks to denote where the multi-model means of CMIP5 and CMIP6 models are statistically different.

The large dots in Figs. 6–7 follow the convention of Fig. 2 from G19 to test whether the mean trend is statistically different from zero, so we prefer to follow the same format to allow comparison with our previous study.

**Was the goal of Fig 2-3 on focusing on the NH only to draw out the differences seen in Fig 1 for JJA in NH?**

Yes. We now clarify in the text (lines 191-192 of the revised manuscript) why we focus on the NH JJA response in Figs. 2–3:

"we further examine the largest difference between CMIP5 and CMIP6 models identified in Fig. 1: the response of the NH JJA Hadley cell edge to 4xCO$_2$ forcing"

**Fig 5: The legends are a little small and repeated. Suggest making larger and only having one (perhaps at the top or bottom of the panel). Might also help to add 'ensemble mean' for the purple line in the top panels of each sub-plot.**

Following the reviewer's suggestions, we have eliminated the repeated legends in Figure 5, enlarged the text of the remaining legend in panel d, and added a legend entry denoting the "multi-reanalysis mean" in the top panel of Figure 5d.

**L238: only 3 out of the models 'greatly exceed' historical and AMIP runs. Suggest mentioning the reanalysis models by name or putting in the clause 'some of the reanalysis greatly exceed'.**

Per the reviewer's request, we have corrected this sentence to be more precise (lines 264-266 of the revised manuscript):

"For the PSI500 metric (Fig. 4, left column), trends from the ERA-Interim, MERRA-2, and JRA-55 reanalyses in the NH and from the ERA-Interim reanalysis in the SH are substantially larger than the trends from the models' control runs and greatly exceed the trends from the historical and AMIP runs of most models (see also G18, G19)."

**Fig 6-7: I can't really tell one model from another. I am not sure this level of detail is helpful as I found these plots a little overwhelming. The ensemble mean bars are also easily lost in the scatter. This is my personal opinion and the authors can change or not change these figures as they wish.**

In Figs. 6 and 7 (also Fig. S3), we have lightened the symbols for the individual models, so that the ensemble mean bars are more easily visible. We agree with the reviewer that there is a lot of information in these figures, but we are updating a similar figure from a prior study (Fig. 2 of Grise et al. 2019). We show the symbols for individual models for two reasons. First, for the common models that performed both CMIP5 and CMIP6 single forcing runs, it allows the reader to see whether the circulation response to a particular forcing notably changed between the CMIP5 and CMIP6 versions of that model. Second, because different models performed single forcing runs in CMIP5 and CMIP6, it allows the reader to assess which models may be contributing to the differences in the CMIP5 and CMIP6 ensemble mean responses to a particular forcing.

**What is happening in Fig c-d bottom panels for the reanalysis between 1990-2000 – is this the PDO?**

The equatorward anomalies in the SH Hadley cell edge in the early 1990s followed by the poleward anomalies in the SH Hadley cell edge in the late 1990s are consistent with the change in phase of the PDO from positive to negative. The AMIP runs of the models also capture this feature to a lesser extent, showing a pause in the poleward SH Hadley cell edge trend in the early 1990s followed by an acceleration of the poleward SH Hadley cell edge trend in the late 1990s. However, this feature is much larger in the reanalyses, likely because concurrent internal atmospheric variability also contributes to large decadal variability in the Hadley cell edge. Unlike coupled atmosphere-ocean variability, the timing of internal atmospheric variability is not necessarily the same in the AMIP runs of the models as in observations, allowing for notable deviations in the decadal variability of the reanalyses and models seen in Fig. 5.

While this feature is interesting, the focus of Section 4 is on the long-term trends, not on the decadal variability in the reanalysis

time series.  For this reason, we choose not to discuss the decadal variability in the observed Hadley cell edge in the paper.

**Suggest starting this section with 'The' so that the 5 and 21 are separated.**

We have changed the title of section 5 to be "Projected Hadley cell expansion over the 21$^{st}$ century" (see line 365 of the revised manuscript).

**The first page would benefit from adding a title of the paper and stating it is the supplementary material (minimum) or adding title page (if you wish).**

According to the ACP manuscript preparation guidelines, "supplements will receive a title page added during the publication process including title ("Supplement of"), authors, and the correspondence email. Therefore, please avoid providing this information in the supplement."  So, per the guidelines, we have not included a title page to the supplement.

**Response to reviewer #3's comments on Grise and Davis (2020)**

We would like to thank the reviewer for taking time to review our manuscript and to provide helpful comments. Based on the reviewer's comments, we have made a number of minor changes and clarifications to the manuscript. Detailed point-by-point responses to all comments are provided below, and original reviewers' comments are provided in bold type.

**General comments: This is a very thorough and thoughtful study on the expansion of the Hadley cell across CMIP5 models/CMIP6 models/reanalyses. It clearly lays out the key similarities and differences between CMIP5 and CMIP6, and sets the differences between the models and reanalysis in a useful context. I have one main comment, and if it is addressed, the manuscript would be suitable for publication.**

We thank the reviewer for their overall positive assessment of our manuscript and efforts to usefully compare CMIP5 models, CMIP6 models, and reanalyses.

**My comment is that the role of the pattern of SST warming for the NH JJA contraction should be discussed. For CMIP5 models, there are significant differences in Hadley cell edge response between the amipFuture and amip4K experiments for the NH JJA season. So it would be helpful if results from the amip4K experiment were shown for comparison. Furthermore, Zhou et al. (2019) argue that an ITCZ shift related to the pattern of enhanced equatorial SST warming drives the subtropical circulation contraction during NH JJA. This is different from the present manuscript, in which it is argued that a general SST warming (pattern not required) forces the equatorial contraction during summer.**

As requested by the reviewer, we have added results from the amip4K experiments to Figure 2c (also to Figure S2b). The reviewer is correct that the sign of the NH Hadley cell edge response changes when the uniform 4K SST warming is used instead of the patterned 4K SST warming. To clarify this, we have added the following text on lines 215-217 of the revised manuscript:

"However, as pointed out by Zhou et al. (2019), the exact pattern of SST warming is critical for capturing the equatorward contraction of the NH JJA Hadley cell edge seen in the abrupt $4xCO_2$ runs. A uniform 4K SST warming would instead result in a poleward expansion of the NH JJA Hadley circulation (Fig. 2c)."

We note that our original text largely followed from Shaw and Voigt (2015)'s results. They concluded that both the amip4K and amipFuture runs contribute to an equatorward contraction of the circulation in the Pacific basin, with the amip4K changes being slightly weaker. A subsequent careful examination of their paper shows that they exclude the western Pacific from their analysis (see red box in their Fig. 1). When averaging over the entire Pacific basin (and thus also in the zonal mean) as we do

here, a poleward expansion of the summertime circulation in the western Pacific basin is sufficient to overwhelm the equatorward contraction of the circulation in the eastern Pacific basin in the amip4K runs, but not in the amipFuture runs. This accounts for the apparent contradiction in the results of Shaw and Voigt (2015), who argue that the amip4K and amipFuture runs contribute to the same sign of the circulation response, and Zhou et al. (2019), who argue that the amip4K and amipFuture runs contribute to different signs of the circulation response.

**1) How do you define the PSI500 Hadley cell edge for NH JJA if PSI never becomes positive in the tropics, and hence a zero-crossing does not exist? In my experience this occurs for some years in certain models.**

The reviewer is correct that the NH JJA Hadley cell edge is undefined during some years. We have added the following text to the methods section (lines 132-136 of the revised manuscript) to clarify this to the reader:

"We note that the NH summertime Hadley circulation is very weak, making it challenging to define the PSI500 metric during some years. We only consider the PSI500 metric from years in which there is a clear crossing of the 500-hPa streamfunction field from positive to negative in the NH subtropics. We consider the PSI500 metric to be undefined if no zero crossing in the streamfunction field occurs or if multiple zero crossings from positive to negative occur within a 20° latitude band ('Lat_Uncertainty = 20' in TropD)."

**2) How exactly is the "response" defined for the abrupt4xCO2 experiment (Fig. 1)? I.e. over what period of the abrupt4xCO2 experiment are you averaging?**

The reviewer must have missed this definition, which is stated in the second sentence of section 3 (lines 155-157): "difference in the Hadley cell edge latitude between its mean position during the last 50 years (years 101-150) of the abrupt $4xCO_2$ run and its mean position in the pre-industrial control run."

We have now also added this definition to the caption of Fig. 1 (lines 699-701) to make sure readers are aware of how the quantity plotted in Fig. 1 is defined.

**References: Zhou, W., S.-P. Xie and D. Yang (2019): Enhanced equatorial warming causes deeptropical contraction and subtropical monsoon shift. Nature Climate Change. 9. 834- 839.**

Thanks. We have added a citation to this paper (lines 663-664).

**List of changes to the manuscript**

Abstract
- Following reviewer #1's suggestion, we have reworded the sentence on lines 15–17.

Section 1
- Following reviewer #2's suggestion, we have reworded the first sentence of the introduction (lines 21-22 of the revised manuscript).
- We have added references to Lucas et al. (2012), Lucas et al. (2014), and Nguyen et al. (2015) in response to reviewer #2's comment.
- Following reviewer #1's suggestion, we have changed "first" to "for example" (line 44) and "second" to "additionally" (line 62).
- In response to reviewer #2's comment, we have added text into the introduction to more fully describe the metrics used to define the edges of the tropics (lines 45–58 of the revised manuscript).

Section 2
- We have rewritten the first paragraph of section 2.1 in response to reviewer #2's comment.
- We have added a description of the amip4K runs, which we now use in response to reviewer #3's suggestion.
- We have removed the details of the reanalysis data sets from the text and placed them in Table 2 (per reviewer #2's suggestion).
- In response to reviewer #3's comment, we have added text to the methods section (lines 132-136 of the revised manuscript) to address how we deal with the Northern Hemisphere summer Hadley cell edge being poorly defined during some years.
- We have eliminated use of the eddy-driven jet metric (per the comments of Reviewers #1 and #2), and replaced it with the USFC metric to identify longitudinal asymmetries in the circulation trends.
- In response to Reviewer #2's comment, we have added text on lines 140-145 addressing the regional meridional overturning cells or local Hadley cell perspective used by some previous studies, explaining why we do not use it in this study.
- In response to Reviewer #2's comment, we had added a paragraph at the end of section 2b to address how we calculate statistical significance (lines 146-151 of the revised manuscript).

Section 3
- We have changed "drastic" to "dramatic" on line 168 in the revised manuscript, per reviewer #1's suggestion.
- We added a citation to a new paper by Zelinka et al. (2020) documenting the higher climate sensitivity in CMIP6 models.
- We have changed "is supported by" to "is consistent with" on line 189 in the revised manuscript, per reviewer #1's suggestion.
- In response to reviewer #2's comment, we now clarify in the text (lines 191-192 of the revised manuscript) why we focus on the NH JJA response in Figs. 2–3.
- As requested by reviewer #3, we have added results from the amip4K experiments to Figure 2c (also to Figure S2b) and describe the results on lines 215-217 of the revised manuscript, including a citation to the Zhou et al. (2019) paper referenced by reviewer #3.
- We have eliminated use of the eddy-driven jet metric (per the comments of Reviewers #1 and #2), and replaced it with the USFC metric to identify longitudinal asymmetries in the circulation trends. Results in Fig. 3 and S2 and

the discussion in section 3 have been updated accordingly.

Section 4

- Per reviewer #2's request, we have corrected the sentence on lines 264-266 to be more precise.
- In response to reviewer #1's confusion, we have added a parenthetical note "(compare orange, black, and red lines in Figs. 5a-b)" to clarify on line 318 in the revised manuscript.

Section 5

- In response to reviewer #2's comment, we have changed the title of section 5 to be "Projected Hadley cell expansion over the 21st century" (see line 365 of the revised manuscript).
- We have changed "when which" to "at which" on line 374 of the revised manuscript, per reviewer #1's suggestion.

Figures and Tables

- Following reviewer #2's suggestion, we have removed Tables S1 and S2 and placed the salient information into a new table (Table 1) in the main text. We have also added the horizontal resolution of each model into the table, as the reviewer requested.
- Following reviewer #2's suggestion, we have added a new table (Table 2) listing the reanalysis data sets used in this study and removed the information from the text in section 2.1.
- Following reviewer #2's suggestion, we have reformatted Fig. 1, Fig. S1, and Fig. S5 in a 1 row-2 column format.
- Per reviewer #3's comment, we have added the definition of the response to the caption of Fig. 1 (lines 699-701) to make sure readers are aware of how the quantity plotted in Fig. 1 is defined.
- As requested by reviewer #3, we have added results from the amip4K experiments to Figure 2c (also to Figure S2b).
- We have eliminated use of the eddy-driven jet metric (per the comments of Reviewers #1 and #2), and replaced it with the USFC metric to identify longitudinal asymmetries in the circulation trends. Results in Fig. 3 and S2 have been updated accordingly.
- Following reviewer #2's suggestions, we have eliminated the repeated legends in Figure 5, enlarged the text of the remaining legend in panel d, and added a legend entry denoting the "multi-reanalysis mean" in the top panel of Figure 5d.
- In response to reviewer #2's comment, in Figs. 6 and 7 (also Fig. S3), we have lightened the symbols for the individual models, so that the ensemble mean bars are more easily visible.

[revised manuscript text omitted]